# Multicellular model of neuroblastoma proposes unconventional therapy based on multiple roles of p53

**Kenneth Y. Wertheim**[1,2,3,4], **Robert Chisholm**[2], **Paul Richmond**[2], **Dawn Walker**[1,2]*

**1** Insigneo Institute for *in Silico* Medicine, University of Sheffield, Sheffield, United Kingdom, **2** School of Computer Science, University of Sheffield, Sheffield, United Kingdom, **3** Centre of Excellence for Data Science, Artificial Intelligence, and Modelling, University of Hull, Kingston upon Hull, United Kingdom, **4** School of Computer Science, University of Hull, Kingston upon Hull, United Kingdom

* d.c.walker@sheffield.ac.uk

**Data Availability Statement:** Model source code is available in the following Github repository: https://github.com/primagesheffield/flamegpu2-

## Abstract

Neuroblastoma is the most common extra-cranial solid tumour in children. Over half of all high-risk cases are expected to succumb to the disease even after chemotherapy, surgery, and immunotherapy. Although the importance of *MYCN* amplification in this disease is indisputable, the mechanistic details remain enigmatic. Here, we present a multicellular model of neuroblastoma comprising a continuous automaton, discrete cell agents, and a centre-based mechanical model, as well as the simulation results we obtained with it. The continuous automaton represents the tumour microenvironment as a grid-like structure, where each voxel is associated with continuous variables such as the oxygen level therein. Each discrete cell agent is defined by several attributes, including its cell cycle position, mutations, gene expression pattern, and more with behaviours such as cell cycling and cell death being stochastically dependent on these attributes. The centre-based mechanical model represents the properties of these agents as physical objects, describing how they repel each other as soft spheres. By implementing a stochastic simulation algorithm on modern GPUs, we simulated the dynamics of over one million neuroblastoma cells over a period of months. Specifically, we set up 1200 heterogeneous tumours and tracked the *MYCN*-amplified clone's dynamics in each, revealed the conditions that favour its growth, and tested its responses to 5000 drug combinations. Our results are in agreement with those reported in the literature and add new insights into how the *MYCN*-amplified clone's reproductive advantage in a tumour, its gene expression profile, the tumour's other clones (with different mutations), and the tumour's microenvironment are inter-related. Based on the results, we formulated a hypothesis, which argues that there are two distinct populations of neuroblastoma cells in the tumour; the p53 protein is pro-survival in one and pro-apoptosis in the other. It follows that alternating between inhibiting MDM2 to restore p53 activity and inhibiting ARF to attenuate p53 activity is a promising, if unorthodox, therapeutic strategy. The multicellular model has the advantages of modularity, high resolution, and scalability, making it a potential foundation for creating digital twins of neuroblastoma patients.

neuroblastoma All supporting data is available in the manuscript and Supplementary Materials.

**Funding:** DW and PR were investigators on the PRIMAGE Project, Grant agreement number 8264941038 awarded by the European Union's Horizon 2020 Research and Innovation Programme. https://cordis.europa.eu/project/id/826494 The funders did not play any role in study design, data collection and analysis, decision to publish, or preparation of the manuscript.

**Competing interests:** The authors have declared that no competing interests exist.

## Author summary

Neuroblastoma is the most common extra-cranial solid tumour in children. Although very low–risk, low-risk, and intermediate-risk cases have good survival rates, most high-risk patients succumb to the disease or relapse despite modern therapies. Neuroblastoma biology is still full of unanswered questions and successful treatment is likely to require a personalised approach. Mathematical and computational modelling could contribute to a better understanding of its origin and nature, as well as the ongoing effort to advance patient-specific therapies, partly because of the potential for integration with other emerging technologies such as multi-omics, medical imaging, data mining, machine learning, and high-performance computing. During the PRIMAGE project, we built, calibrated, and validated, to the best of our knowledge, the first multicellular computational model of neuroblastoma. As we reported in an earlier paper, this was integrated into a multi-scale orchestrated computational framework. Here, we describe how we used the multicellular computational model as a standalone component, to gain new insights into neuroblastoma. Armed with it, we carried out large-scale computer simulations involving over a million independent cell agents in the most expensive case and 5000 hypothetical drug combinations. Our results confirm and add new insights into the non-linear dynamics spanning the intracellular, intercellular, and microenvironmental scales of neuroblastoma. They also indicate that an unconventional therapy targeting a frequently overlooked role of *p53* (DNA repair) is promising. On the basis of these results, we believe that our model will be useful for the *in silico* medicine community. We hope that it will one day become the foundation of a virtual neuroblastoma.

## 1 Introduction

Neuroblastoma is the most common extra-cranial solid tumour in children [3]: the median age of diagnosis is 573 days [4]. Its origin is related to the stem cells and progenitor cells in the neural crest, which is a transient structure in the human embryo [5–7]. In the normal course of developmental events, these multipotent cells migrate and differentiate into diverse cell lineages, including the sympathoadrenal lineage, which generates the sympathetic nervous system. If the *MYCN* oncogene is amplified and the *ALK* oncogene acquires activating mutations in this lineage, the cells therein will become neuroblastoma cells and form a malignant solid tumour instead [6, 8].

Neuroblastoma is a clinically heterogeneous disease. The International Neuroblastoma Risk Group (INRG) classification system [9] stages neuroblastoma based on the patient's age, image-defined risk factors, and tumour metastasis. Within the same system, risk stratification relies on more criteria, including the patient's age, stage, histology, and additional clinical and molecular parameters. Although the very low–risk, low-risk, and intermediate-risk groups within this system have a five-year overall survival rate above 70%, over half of the high-risk patients are expected to succumb to the disease or relapse even after multi-modal therapy, whereupon survival for more than three years is rare [10]. This multi-modal therapy—the current standard of care for high-risk neuroblastoma—combines chemotherapy, surgery, radiation therapy with stem cell rescue, and immunotherapy [11].

Neuroblastoma's biological features mirror its clinical heterogeneity and it is known to display intra-tumour heterogeneity [12]. In one study [12], pangenomic techniques identified heterogeneous genomic profiles in terms of chromosomal aberrations in almost 40% of 58

high-risk tumours. Furthermore, the patients with these heterogeneous tumours were found to have a significantly better survival rate, especially when the *MYCN* gene was not amplified. In addition, the heterogeneity of *MYCN* amplification in a tumour is known to influence its clinical features. The study supporting this claim [13] compared tumours without this mutation, tumours wherein *MYCN* was homogeneously amplified (more than 20% of cancer cells exhibited *MYCN* amplification), and tumours wherein it was heterogeneously amplified. The patients with heterogeneously *MYCN*-amplified tumours had higher lactate dehydrogenase (LDH) levels (predictor of poor clinical outcomes) than those with homogeneously *MYCN*-amplified tumours. Different biological features were also found in the three groups of patients. For example, the patients with heterogeneously *MYCN*-amplified tumours were less likely to have unfavourable features such as loss of heterozygosity (LOH) at chromosome arm 1p. On the other hand, the two groups with *MYCN* amplification went on to attain similar event-free and overall survival rates, even though they both suffered worse outcomes than the patients without this mutation.

With respect to heterogeneously *MYCN*-amplified tumours only, the genomic background (chromosomal aberrations) of such a tumour, the reproductive advantage of the *MYCN*-amplified clone therein, and its disease outcome are known to be related in a counter-intuitive manner [14, 15]. Specifically, numerical chromosomal aberrations favour *MYCN*-amplified clones, but are associated with better survival rates, while segmental chromosomal aberrations constrain the clone, but lead to aggressive disease progression. Consistently, one of the studies observed that a more dominant *MYCN*-amplified clone did not always result in a worse clinical outcome [15].

Another defining feature of neuroblastoma is a non-linear relationship between the *MYCN*-amplified clone's reproductive advantage in a tumour, the tumour's other mutations, and its gene expression profile. First, MYCN favours both proliferation and apoptosis [16]. The balance between these opposing mechanisms depends on the products of genes such as *Bcl2* and *p53*. In particular, MYCN and the ARF/MDM2/p53 axis are connected in an elaborate gene regulatory network [17], a plausible explanation for why neuroblastoma's clinical outcome is non-linearly related to *MYCN* expression [18]. Second, mutations other than *MYCN* amplification also drive disease progression. High-risk neuroblastoma tumours frequently suffer from *TERT* rearrangement and *ATRX* inactivation [19]. They are three almost mutually exclusive means of activating telomere repair. According to a mechanistic classification of clinical phenotypes for neuroblastoma [20], disease progression is likely when one of them is present. Independently of these three mutations, when the MAPK/RAS pathway is activated or the p53 pathway is inactivated, the classification scheme predicts a poor disease outcome.

Mathematical and computational models of cancers are useful for examining patient-specific therapies, not least because of the potential for integration with other emerging technologies such as multi-omics data, medical images, data mining, machine learning, and high-performance computing [21]. Systems modelling is considered useful for neuroblastoma research and management by the scientific community [22]. Currently, the community—similar to the case in cancer research in general—has the ambition of personalising diagnosis, prognosis, and treatment [23] with the help of multi-scale modelling and imaging biomarkers [1, 24–28], as well as patient-derived preclinical models [29, 30] and liquid biopsies [31]. This paper presents our contributions to this goal. To complement the subcellular models reported in the literature [32–35], we developed—to the best of our knowledge—the first multicellular computational model of neuroblastoma during the PRIMAGE project [1], comprising three major components: a continuous automaton, discrete cell agents, and a centre-based mechanical model. In a recently published paper [2], our model is presented as a part of a multi-scale

orchestrated computational framework. In this paper, we report what we accomplished with the multicellular model as a standalone component, outside the multi-scale framework. Model parameters were calibrated with respect to experimental and clinical data. We then carried out simulations of *MYCN*-amplified clones' dynamics in a broad range of heterogeneous tumours, revealing the conditions that promote their growth, and tested 5000 hypothetical combination therapies inhibiting 20 drug targets. The most computationally expensive experiment involved simulating over a million independent cell agents. These simulations could not be implemented using traditional processors, single or parallel [36]. As a result, it was necessary to utilise modern graphics processing units (GPUs) with a highly parallel architecture, such as NVIDIA A100 Tensor Core GPU Architecture (white paper: link). FLAME GPU 2 [37], an agent-based simulation framework which targets NVIDIA GPUs, abstracts the complexity of GPU parallelism away from a modeller, allowing them to exploit highly efficient CUDA-based model code without knowledge of this specialised programming language. Thanks to this layer of abstraction, the modellers involved in this project could just implement the model using C++. Our results confirm and add new insights into the non-linear relationship between the *MYCN*-amplified clone's reproductive advantage in a tumour, the tumour's other clones (with other mutations), and its gene expression profile. Our insights broaden the scope of this relationship to include the *MYCN*-amplified clone's gene expression profile and the tumour's microenvironment. The results of the *in silico* drug trial reveal potentially promising drug combinations.

## 2 Results

The architecture of our multicellular model of neuroblastoma is illustrated in Fig 1. There are two populations of autonomous discrete agents representing neuroblastoma and Schwann cells (healthy cells residing in the tumour [38]), which stimulate and inhibit each other as illustrated in Fig 1A. In this manuscript, within the context of the multicellular model, the words 'cell' and 'agent' are used interchangeably. They reside in a continuous automaton describing the tumour's microenvironment as a grid-like structure, illustrated in Fig 1B, where each voxel is associated with continuous variables such as the oxygen level therein. Each discrete cell agent is defined by a number of attributes, including its cell cycle position, mutations, gene expression pattern, and more. Its behaviour is stochastically dependent on these attributes and includes, amongst others, cell cycling and cell death. Mathematically, its stochastic behaviour is modelled by Bernoulli trials with conditional probabilities, as illustrated in Fig 1C. Finally, there is a centre-based mechanical model of cell-cell repulsion: Fig 1D. It describes the properties of these agents as physical objects, describing how they repel each other as soft spheres. Fig 1E defines the population structure wherein a neuroblastoma cell agent belongs to one of four clones, which are each divided into six smaller clones (or subclones in this exposition).

A simulation algorithm was developed to integrate these model components in an iterative process (illustrated in Fig 2). Within this process, the steps listed in Fig 2 are arranged in a deterministic order, but the cellular decisions are stochastic. After calibrating the model with mostly *in vitro* data, as illustrated in Fig 3 and Table 1, and refining the parameters for *in vivo* scenarios, we simulated the dynamics of clonal competition within 1200 heterogeneous tumours. Then, we compared *MYCN*-amplified clones with different gene expression profiles, fractional compositions, and microenvironments. Finally, we conducted an *in silico* trial of 5000 hypothetical drug combinations targeting the 20 gene products modelled within the neuroblastoma cell agents. Please refer to section 5 for further information relating to this process, with full details included in S1–S4 Text files. In this section, we will present the results generated from each set of simulations.

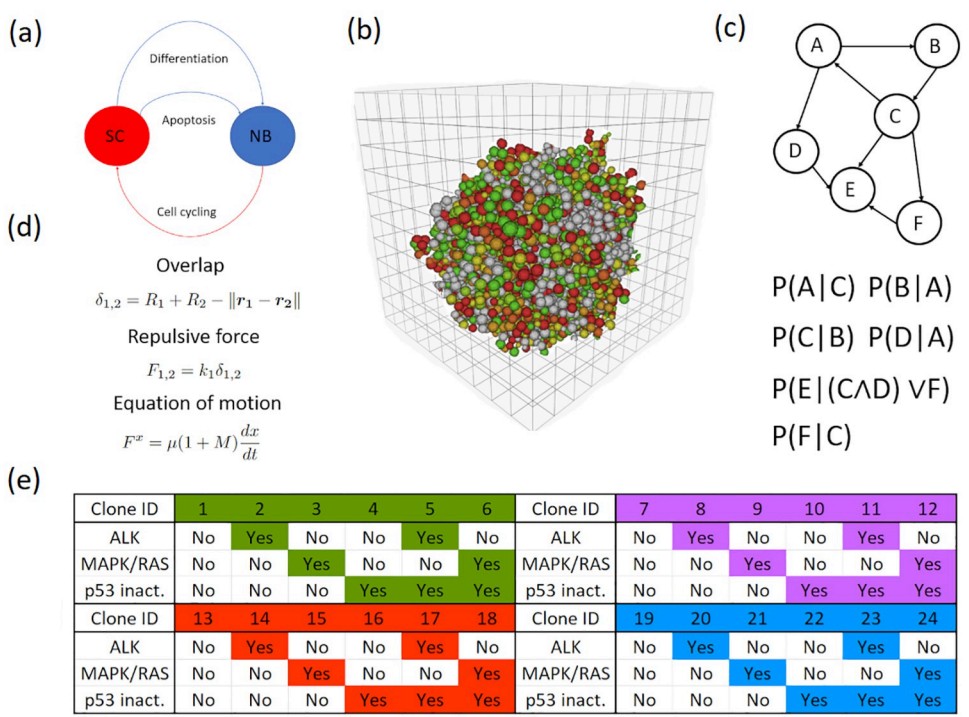

**Fig 1. Model structure.** (A) There are two agent populations: neuroblastoma and Schwann cell agents. They communicate using intercellular juxtacrine (contact-dependent) and paracrine (diffusive) signals, such as apoptotic signals. (B) The tumour's microenvironment is represented by a continuous automaton, which comprises voxels populated by the cell agents. (C) The intracellular mechanisms mediating cell cycling and cell death in response to the microenvironment and intercellular interactions are represented by Bernoulli trials with conditional probabilities (full details provided in S1 Text). (D) Cell-cell overlap is resolved by a centre-based mechanical model (also known as an off-lattice soft sphere model), wherein cells repel each other like soft spheres. (E) A neuroblastoma cell agent belongs to one of four clones: wild-type (green), *MYCN*-amplified (magenta), *TERT*-rearranged (red), and *ATRX*-inactivated (blue). Each clone comprises six smaller clones (or subclones) with different combinations of the following: *ALK* activation/amplification, other mutations activating the MAPK/RAS signalling pathway, and *p53* inactivation. At the highest resolution, a neuroblastoma cell agent's clonal identity is defined in terms of the subclone to which it belongs. For the sake of simplicity, it has a clone ID.

## 2.1 Clonal competition within heterogeneous tumours

We conducted an experiment to compare the impact of clonal composition on a tumour's fate with that of macroscopic conditions (such as hypoxia). The details are provided in subsubsection 5.6.1. Following the literature [20, 43], we classified the 1200 ensemble outcomes of the first set of simulations into three categories: regression, differentiation, and progression. If a run finished without any living neuroblastoma cells, it was considered a regressed tumour. If a run finished with just living neuroblastoma cells that were at least 90% differentiated on average, it was considered a differentiated tumour. If a run finished with just living neuroblastoma cells that were less than 90% differentiated on average, it was considered a progressing tumour. Only cell counts, not tumour volumes, were considered in this classification step. An ensemble was categorised only if the majority of its runs had ended in one of the three outcomes. For example, if more than five runs in an ensemble had led to regressed tumours, the ensemble would be placed in the category of regression. Based on the ensemble outcomes, 45 virtual tumours regressed, none differentiated, and 1155 progressed. To address the level of uncertainty surrounding this classification system, we assessed the distribution of runs over the three outcomes in each ensemble. In 1154 ensembles, every run produced a progressing

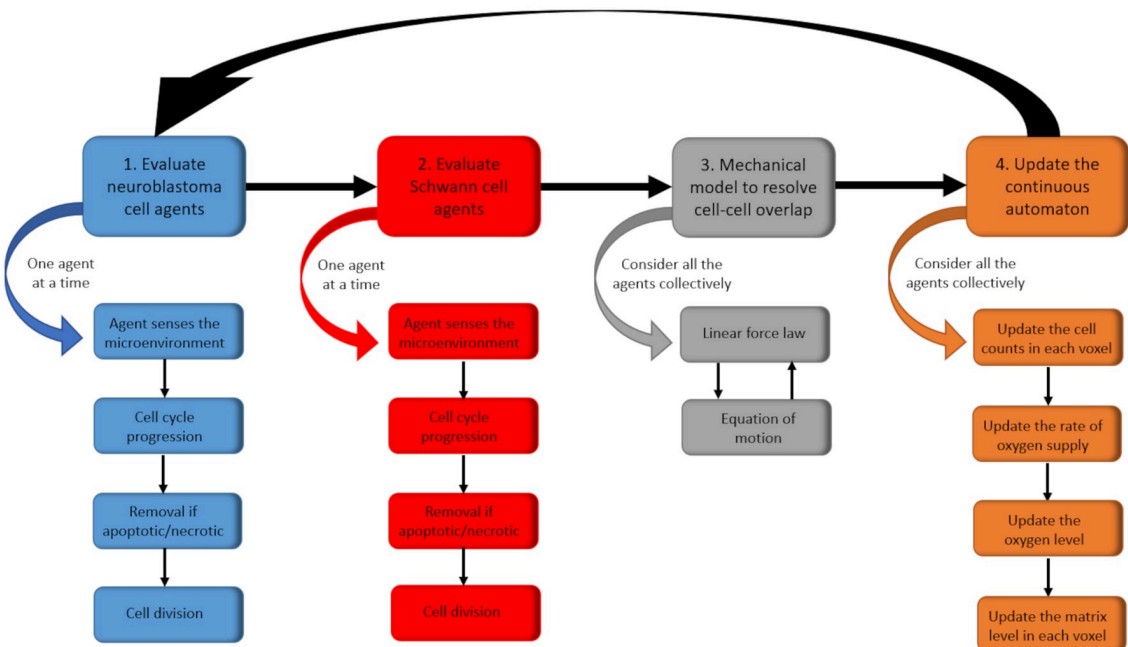

**Fig 2. Sequence of events in one realisation of the stochastic process.** At the start of each time step (one hour), any neuroblastoma cell agents are sequentially evaluated. Each agent senses and responds to its microenvironment by updating its attributes. Then, it attempts to progress in the cell cycle. If it is apoptotic or necrotic, it may get removed from the system. Otherwise, it may divide into two daughter agents. Schwann cell agents are then updated similarly. All agents are then considered collectively to minimise the total overlap in the system, thus updating their spatial coordinates. This involves solving a centre-based mechanical model by applying Euler's method iteratively. Each step finishes with an update of the continuous automaton. This accounting step does not change the agents' attributes; it simply records their latest spatial distributions, any changes in the supply rate of oxygen and its abundance, and matrix production.

tumour. In 42 ensembles, every single run produced a regressed tumour. The remaining four ensembles were found to contain runs producing one progressing and nine regressed tumours; three progressing and seven regressed tumours; one progressing and nine regressed tumours; and six progressing and four regressed tumours. The runs converged to their ensemble outcomes with just several exceptions. This convergence, as well as the high number (1196 out of 1200) of ensembles with unique classifications, is reassuring because, on this basis, the presented ensemble outcomes are unlikely to be the byproducts of how we classified the runs rather than biologically meaningful results. The fact that not a single run produced a differentiated tumour is also reassuring because we were interested in malignant tumours and did not evaluate any virtual tumours with more Schwann cell agents than neuroblastoma cell agents. That differentiation did not occur as an emergent phenomenon during the simulations is consistent with the consensus that spontaneous regression or differentiation typically occurs in benign cases [44].

Figs 4, 5 and 6 collectively explore the input space of the first set of simulations. Fig 4B indicates that, as expected, hypoxia led to regression. On the other hand, the initial clonal composition of a virtual tumour had very little impact on its ensemble outcome (Figs 5 and 6). Recall that a clonal composition refers to the distribution of neuroblastoma cell agents within the population structure defined in Fig 1E. The fractional composition of a clone or subclone is simply the number of living neuroblastoma cell agents belonging to that clone or subclone divided by the total number of living neuroblastoma cell agents in a virtual tumour. It does not

**Fig 3. Calibration pipeline.** We generated 3000 near-random combinations of 20 unconstrained parameters by Latin hypercube sampling and systematically eliminated unrealistic sets of parameters in a series of calibration studies. In the first study, they were assessed by their ability to reproduce neuroblastoma's *in vitro* growth kinetics [39]. In the second study, their ability to mimic its hypoxic response was evaluated [40]. The third study was designed based on the regulatory dynamics between neuroblastoma and Schwann cells, observed *in vitro*[41]. The fourth study selected the parametric combinations that reproduced the clinical outcomes associated with different histological categories [42]. The last two studies were designed to reproduce the clinical outcomes associated with different mutations [20].

refer to the tumour volume occupied by the clone or subclone. As described in subsubsection 5.6.1, each of these 1200 virtual tumours was given a random initial clonal composition. In the 24 violin plots in Figs 5A, 5D, 6A and 6D, the 1200 medians are very similar (around 0.03). Their counterparts in Figs 5B, 5E, 6B and 6E, which are pertinent to the 45 regressed virtual

**Table 1. Summary of the calibration process.** We generated 3000 near-random combinations (column labelled 'Sets' below) of 20 unconstrained parameters by Latin hypercube sampling and systematically eliminated unrealistic sets of parameters in a series of calibration studies. In each study, we populated our computational model with the remaining parametric sets before attempting to reproduce an experimentally observed phenomenon.

| Study | Experimentally observed phenomenon | Sets | Dataset |
|---|---|---|---|
| 1 | Growth kinetics of a neuroblastoma cell line (IMR 32) *in vitro* | 3000 | [39] |
| 2 | Hypoxic response of avascular spheroids *in vitro* | 1000 | [40] |
| 3 | Interactions between neuroblastoma and Schwann cells in mouse xenograft models | 50 | [41] |
| 4 | Tumour histological category's influence on clinical outcome | 10 | [42] |
| 5 | Tumour mutation profile's influence on clinical outcome | 4 | [20] |
| 6 | Tumour mutation profile's influence on clinical outcome (*MYCN*-amplified only) | 3 | [20] |

## Initial macroscopic properties of heterogeneous tumours

**Fig 4. Violin plots illustrating the initial macroscopic properties of 1200 heterogeneous virtual tumours, namely each tumour's dimensionless oxygen level (O2) and the fraction of its total cell population assigned to be Schwann cells (SC fraction).** (A) Each violin plot contains 1200 data points (all cases), one for each virtual tumour. (B) Each violin plot contains 45 data points (regressed cases only), one for each regressed tumour. (C) Each violin plot contains 1155 data points (progressing cases only), one for each progressing tumour.

tumours, present no evidence of a significantly more dominant clone (highest median is less than 0.05). Finally, in Figs 5C, 5F, 6C and 6F, the 24 violin plots summarising the progressing cases are similar to those summarising the entire set of simulations. The lack of any significant variations in Figs 5 and 6 indicates that the model's ensemble outcome is generally not very sensitive to its initial clonal composition. For each subclone, we resampled from its 1200 initial fractional compositions to build two confidence intervals. First, we resampled the sample 12000 times, taking 45 values with replacement and recording their medium on each occasion. Using the 12000 mediums, we built a 95% confidence interval for the subclone's medium initial fractional composition. Second, we repeated the procedures with a resample size of 1155 values to build another 95% confidence interval. We compared the subclone's medium initial fractional composition in the 45 virtual tumours that ended in regression with the first confidence interval and the medium in the 1155 virtual tumours that ended in progression with the second confidence interval. The results are presented in Fig 7. Only one point estimate of a medium (subclone four in the regressed virtual tumours) falls outside its confidence interval. Considering the probability that a result that extreme occurs by chance is 5% and we calculated 48 confidence intervals, it is reasonable to dismiss that point estimate as a chance event rather than a significant variation. We conclude that the model's ensemble outcome is generally not very sensitive to its initial clonal composition.

Fig 8 presents the outcomes of the 1155 progressing virtual tumours only. Fig 8A–8D presents the final states of the four groups of 1155 clones. A clone's final enrichment refers to the ratio of its final fractional composition to its initial fractional composition. Fig 8E–8H presents the fractional compositions of the six subclones in each clone in a virtual tumour at the end of its corresponding simulation. There are two key outcomes suggested within this figure.

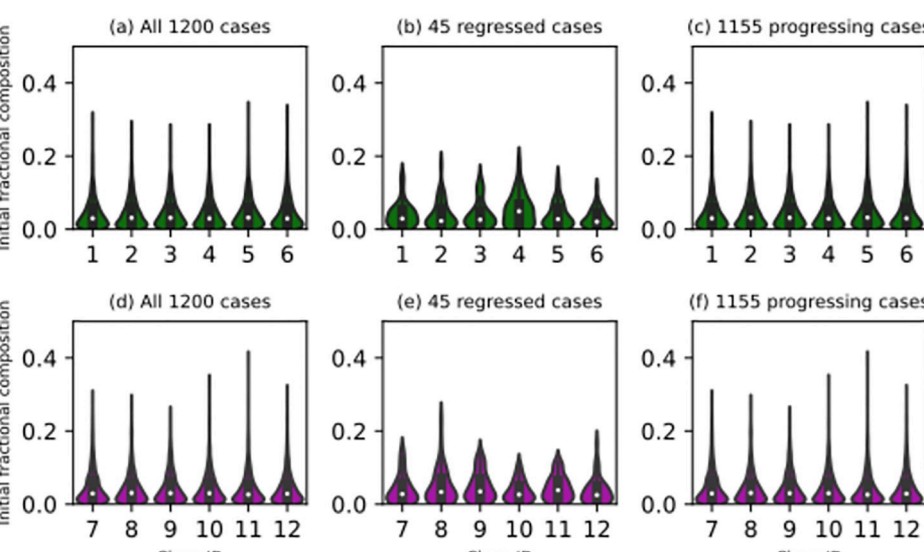

**Fig 5. Violin plots illustrating the initial fractional compositions of the wild-type (WT, green) and *MYCN*-magnified (MA, magenta) clones in 1200 heterogeneous virtual tumours.** The fractional composition of a clone or subclone is simply the number of living neuroblastoma cell agents inside that clone or subclone divided by the total number of living neuroblastoma cell agents in the virtual tumour containing the clone or subclone. (A) and (D) Each violin plot contains 1200 data points (all 1200 cases), one for each virtual tumour. (B) and (E) Each violin plot contains 45 data points (regressed cases only), one for each regressed tumour. (C) and (F) Each violin plot contains 1155 data points (progressing cases only), one for each progressing tumour.

First, as evident in Fig 8B, the 1155 progressing virtual tumours' *MYCN*-amplified clones all shrank, while the other three groups of clones expanded in most cases. As explained in sub-section 5.7, a one-tailed paired sample t-test was performed on the final enrichment levels associated with the wild-type and *MYCN*-amplified clones: Fig 8A and 8B. With the null hypothesis that the true means of the two populations (normal distributions) are the same, we obtained a p-value less than $2.2 \times 10^{-16}$, supporting the hypothesis that the two datasets are statistically different. In order to test whether the three datasets associated with the wild-type, *TERT*-rearranged, and *ATRX*-inactivated clones—Fig 8A, 8C and 8D—are statistically different, we performed an ANOVA analysis as described in subsection 5.7. With the null hypothesis that the three underlying populations have the same mean, we obtained p-values of 0.2046 and 0.72 respectively, finding little evidence that the three datasets are statistically different. Therefore, it is reasonable to claim that in a progressing virtual tumour, the wild-type, *TERT*-rearranged, and *ATRX*-inactivated clones behaved identically on average.

Second, there is a unifying theme across the datasets pertaining to the three groups of expanding clones. Fig 8E shows that within the wild-type clones, three (clone ID: 1, 2, and 3) of the six groups of subclones dominated the remaining three when the simulations ended. Fig 8G and 8H indicates the same trend in the TERT-rearranged (dominant clone ID: 13, 14, and 15) and ATRX-inactivated clones (dominant clone ID: 19, 20, and 21). In each case, the three competitive groups of subclones did not have an inactivated p53 pathway. A fitter clone or subclone, contained more of the living neuroblastoma cell agents in a virtual tumour at the end of the relevant simulation than a less fit one. By this definition, the fittest group had an activated/amplified *ALK* gene (clone ID: 2, 14, and 20).

## Initial fractional compositions in heterogeneous tumours (TR and AI)

**Fig 6. Violin plots illustrating the initial fractional compositions of the *TERT*-rearranged (TR, red) and *ATRX*-inactivated (AI, blue) clones in 1200 heterogeneous virtual tumours.** The fractional composition of a clone or subclone is simply the number of living neuroblastoma cell agents inside that clone or subclone divided by the total number of living neuroblastoma cell agents in the virtual tumour containing the clone or subclone. (A) and (D) Each violin plot contains 1200 data points (all 1200 cases), one for each virtual tumour. (B) and (E) Each violin plot contains 45 data points (regressed cases only), one for each regressed tumour. (C) and (F) Each violin plot contains 1155 data points (progressing cases only), one for each progressing tumour.

|  | ID1 | ID2 | ID3 | ID4 | ID5 | ID6 |  |  | ID1 | ID2 | ID3 | ID4 | ID5 | ID6 |
|---|---|---|---|---|---|---|---|---|---|---|---|---|---|---|
| Medium (R) | 0.029 | 0.023 | 0.026 | **0.049** | 0.028 | 0.020 | | Medium (O) | 0.030 | 0.031 | 0.031 | 0.029 | 0.032 | 0.030 |
| CI LB (R) | 0.020 | 0.018 | 0.020 | **0.019** | 0.021 | 0.019 | | CI LB (O) | 0.028 | 0.028 | 0.029 | 0.027 | 0.030 | 0.026 |
| CI UB (R) | 0.043 | 0.045 | 0.045 | **0.043** | 0.046 | 0.044 | | CI UB (O) | 0.032 | 0.034 | 0.034 | 0.032 | 0.034 | 0.032 |
|  | ID7 | ID8 | ID9 | ID10 | ID11 | ID12 |  |  | ID7 | ID8 | ID9 | ID10 | ID11 | ID12 |
| Medium (R) | 0.028 | 0.033 | 0.035 | 0.026 | 0.038 | 0.025 | | Medium (O) | 0.029 | 0.031 | 0.030 | 0.030 | 0.026 | 0.029 |
| CI LB (R) | 0.018 | 0.019 | 0.019 | 0.018 | 0.017 | 0.018 | | CI LB (O) | 0.027 | 0.028 | 0.028 | 0.027 | 0.024 | 0.026 |
| CI UB (R) | 0.045 | 0.044 | 0.044 | 0.044 | 0.041 | 0.042 | | CI UB (O) | 0.031 | 0.033 | 0.033 | 0.033 | 0.029 | 0.031 |
|  | ID13 | ID14 | ID15 | ID16 | ID17 | ID18 |  |  | ID13 | ID14 | ID15 | ID16 | ID17 | ID18 |
| Medium (R) | 0.029 | 0.030 | 0.042 | 0.038 | 0.028 | 0.031 | | Medium (O) | 0.030 | 0.032 | 0.029 | 0.028 | 0.032 | 0.029 |
| CI LB (R) | 0.020 | 0.019 | 0.019 | 0.018 | 0.019 | 0.018 | | CI LB (O) | 0.028 | 0.029 | 0.027 | 0.027 | 0.029 | 0.027 |
| CI UB (R) | 0.044 | 0.046 | 0.044 | 0.041 | 0.045 | 0.043 | | CI UB (O) | 0.033 | 0.035 | 0.032 | 0.031 | 0.034 | 0.032 |
|  | ID19 | ID20 | ID21 | ID22 | ID23 | ID24 |  |  | ID19 | ID20 | ID21 | ID22 | ID23 | ID24 |
| Medium (R) | 0.032 | 0.024 | 0.031 | 0.022 | 0.028 | 0.028 | | Medium (O) | 0.031 | 0.030 | 0.029 | 0.030 | 0.029 | 0.030 |
| CI LB (R) | 0.020 | 0.019 | 0.019 | 0.020 | 0.018 | 0.019 | | CI LB (O) | 0.029 | 0.028 | 0.026 | 0.027 | 0.026 | 0.028 |
| CI UB (R) | 0.044 | 0.045 | 0.044 | 0.044 | 0.042 | 0.045 | | CI UB (O) | 0.033 | 0.032 | 0.032 | 0.033 | 0.032 | 0.032 |

**Fig 7. Two sets of 95% confidence intervals for 45 regressed and 1155 progressing virtual tumours' medium initial clonal fractional compositions.** The six columns on the left record the 45 regressed virtual tumours' mediums, the lower bounds of their confidence intervals (CI LB), and the upper bounds (CI UB). The six columns on the right record the same statistics for the 1155 progressing virtual tumours. Subclones 1, 2, 3, 4, 5, and 6 constitute the wild-type (green) clone. Subclones 7, 8, 9, 10, 11, and 12 constitute the *MYCN*-amplified (magenta) clone. Subclones 13, 14, 15, 16, 17, and 18 constitute the *TERT*-rearranged (red) clone. Subclones 19, 20, 21, 22, 23, and 24 constitute the *ATRX*-inactivated (blue) clone. At the highest resolution, a neuroblastoma cell agent's clonal identity is defined in terms of the subclone to which it belongs. For the sake of simplicity, it has a clone ID. For each subclone, we resampled from its 1200 initial fractional compositions to build two confidence intervals. First, we resampled the sample 12000 times, taking 45 values with replacement and recording their medium on each occasion. Using the 12000 mediums, we built a 95% confidence interval for the subclone's medium initial fractional composition. Second, we repeated the procedures with a resample size of 1155 values to build another 95% confidence interval.

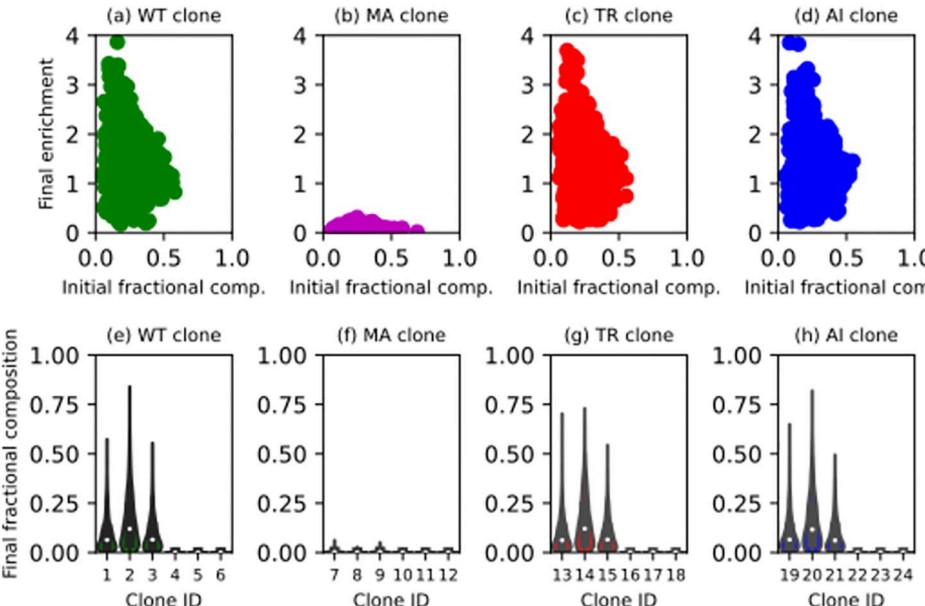

**Fig 8. Outcomes of clonal competition in 1155 progressing heterogeneous virtual tumours.** (A)–(D) The four scatter plots pertain to four groups of clones: wild-type (WT, green), *MYCN*-amplified (MA, magenta), *TERT*-rearranged (TR, red), and *ATRX*-inactivated (AI, blue). Each point on a scatter plot relates a clone's initial fractional composition to its enrichment at the end of the corresponding simulation: its final fractional composition divided by its initial fractional composition. The four clones' fractional compositions in each virtual tumour, both initial and final, always add up to one. (E)–(H) The violin plots present the final fractional compositions of the six subclones in each clone. The 24 fractional compositions of every virtual tumour always add up to one.

## 2.2 MA clone's sensitivity to its gene expression profile

Due to the unexpected behaviour demonstrated by the MA clones in the simulations described in subsection 2.1, further simulations were run to find out an MA clone's sensitivity to its intracellular conditions (expression levels of four genes and one pathway). The experimental details are provided in subsubsection 5.6.2. The violin plots in Fig 9A illustrate the 283 gene expression profiles of the expanding *MYCN*-amplified clones in the second set of simulations. It is clear that the gene expression level of *p73* is upregulated in these profiles. It is also clear that one expression level (MR1) of the MAPK/RAS signalling pathway is upregulated relative to the other (MR0). However, MR1 was set to be higher than MR0 to reflect the influence of *MYCN* amplification on the pathway, so the difference should be considered a hypothesis, not a result. A principal component analysis of this six-dimensional dataset resulted in Fig 9B. The first principal component can explain around 25% of the variance in the dataset (283 profiles) only. The dataset has six dimensions (features). If the examples were evenly distributed in this six-dimensional feature space, a principal component analysis would give a first principal component accounting for around 17% (1/6) of the variance in this hypothetical dataset. In other words, the actual first principal component is less than 1.5 times more explanatory than an observable would be with respect to white noise. Since a principal component is a latent feature, an underlying variable that is not directly recorded (observable) in a dataset but can be inferred from it to explain the variance therein, we conclude from our first (most explanatory) principal component's low predictive power that the six observables (gene expression levels) influence a *MYCN*-amplified clone independently.

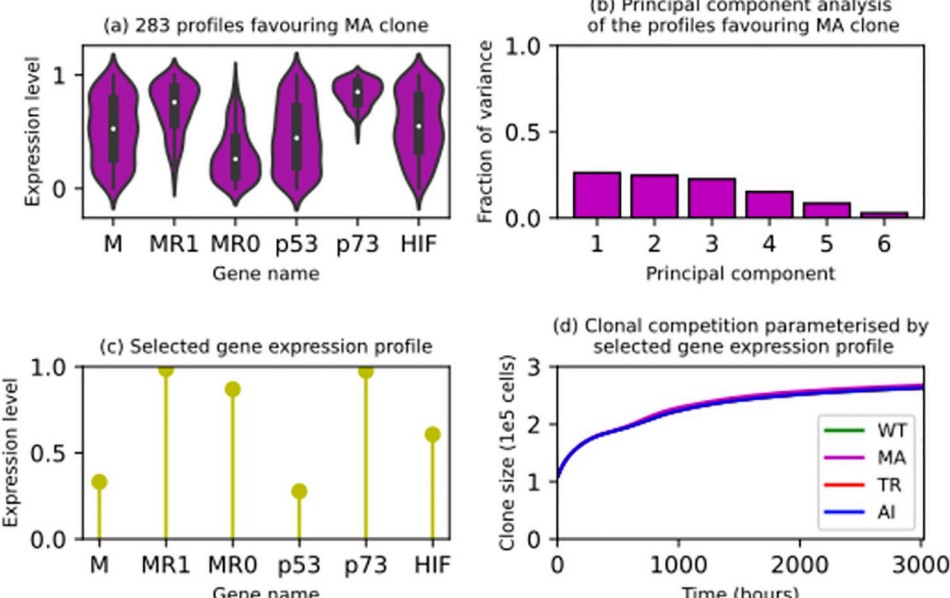

**Fig 9. Sensitivity analysis evaluating 1000 *MYCN*-amplified (MA) clones with different gene expression profiles.**
(A) The violin plots present how the expression levels of each gene are distributed over the profiles that allowed their corresponding *MYCN*-amplified (MA) clones to expand in the simulations. The genes are *MYCN* (M), the genes encoding the MAPK/RAS signalling pathway (MR1 and MR0), *p53*, *p73*, and *HIF*. MR1 denotes enhanced MAPK/RAS signalling in a *MYCN*-amplified neuroblastoma cell agent. MR0 denotes enhanced MAPK/RAS signalling in a neuroblastoma cell agent whose *MYCN* is not amplified. (B) Each bar quantifies the amount of variance in these gene expression profiles that can be explained by a particular principal component. (C) The lollipop chart presents the gene expression profile that parameterised the simulation that matched what is known [20] most closely. In this simulation, the three mutated clones dominated the wild-type clone and in each clone, the subclones with mutations in *p53* and the genes encoding the MAPK/RAS signalling pathway dominated their peers. Furthermore, the number of living neuroblastoma cells in each clone increased during the simulation: the clones expanded. (D) The four time series confirm that, in the simulation parameterised by the gene expression profile presented in (C), all four clones (wild-type or WT, green; *MYCN*-amplified or MA, magenta; *TERT*-rearranged or TR, red; and *ATRX*-inactivated or AI, blue) expanded.

As detailed in subsection 5.7, one particular gene expression profile was selected for the remaining studies. The lollipop chart in Fig 9C illustrates the six expression levels constituting this singular gene expression profile. Compared to the other profiles, this profile parameterised a simulation that progressed in accordance with what is known [20]. In the corresponding virtual tumour, the three mutated clones dominated the wild-type clone. The subclones with mutations in *p53* and the genes encoding the MAPK/RAS signalling pathway dominated their peers in each clone. The number of living neuroblastoma cells in each clone also increased in the simulation. Fig 9D presents the population dynamics in this simulation. Since all four clones expanded, this gene expression profile supports a tumour's growth regardless of the mutations affecting the neuroblastoma cells in it. On average (10 runs), the wild-type, *MYCN*-amplified, *TERT*-rearranged, and *ATRX*-inactivated clones contained approximately $2.63 \times 10^5$ (262714–263672), $2.68 \times 10^5$ (266953–268496), $2.63 \times 10^5$ (262968–264037), and $2.63 \times 10^5$ (262572–264310) living neuroblastoma cell agents when this simulation finished. Although the four trajectories in Fig 9D look identical, the mutated clones did outgrow the wild-type clone during the simulation and the *MYCN*-amplified clone contained more living neuroblastoma cells than any of the other three clones when the simulation ended. As the differences are minute, their biological relevance is questionable, but the gene expression profile

presented in Fig 9C is certainly consistent with what is known about a malignant tumour, as evidenced and explained in subsection 5.7. To strengthen this analysis, statistical tests were performed as described in subsection 5.7. First, a one-tailed paired sample t-test was performed on the 10 final *MYCN*-amplified clone sizes and the 10 final wild-type clone sizes. With the null hypothesis that the true means of the two populations (normal distributions) are the same, we obtained a p-value less than $1.75 \times 10^{-10}$, supporting the hypothesis that the two datasets are statistically different: that on average, the *MYCN*-amplified clone did outgrow the wild-type clone in the simulation. Applying this analysis to the final *TERT*-rearranged and *ATRX*-inactivated clone sizes relative to the final wild-type clone sizes, we obtained p-values of 0.0305 and 0.1374 respectively. The hypothesis that the *TERT*-rearranged clone outgrew the wild-type clone is also statistically significant, but the equivalent hypothesis about the *ATRX*-inactivated clone is statistically less significant. Then, an ANOVA analysis was performed on the final sizes of the three mutated clones, leading to two p-values smaller than $2.2 \times 10^{-16}$ and $2.2 \times 10^{-10}$ respectively, supporting the hypothesis that the three mutated clones achieved significantly different outcomes in the simulation.

## 2.3 MA clone's sensitivity to its fractional composition and microenvironment

We carried out another experiment about the *MYCN*-amplified clone to test if a numerical advantage translates to a reproductive advantage. The experimental details are provided in subsubsection 5.6.3. In the third set of simulations, regardless of the initial oxygen level used in a simulation, the *MYCN*-amplified clone therein was only slightly more dominant in terms of its fractional composition (less than one percent) on average at the end of the simulation than it was at the start. In the two virtual tumours with 25% of their initial neuroblastoma cell agents in their respective *MYCN*-amplified clones, the fractional composition went up to 25.2% at the end of each simulation. 50% initially, 50.3% at the end; 75% initially, 75.2% at the end. These unexciting results are not visualised. A more dominant *MYCN*-amplified clone (numerical advantage) was not necessarily a fitter clone (reproductive advantage) in these simulations.

These results were also used to test the model's robustness. The final fractional compositions of the subclones that remained when the 1000 runs (each of the 10 virtual tumours was simulated by implementing it 100 times) ended were used to build confidence intervals as explained in subsection 5.7. Each virtual tumour was allocated a unique combination of initial clonal composition and oxygen level. The model was implemented 100 times (100 runs) and the final fractional compositions of the subclones that remained when a run ended were recorded, leading to six or 12 datasets corresponding to the six or 12 subclones that remained in the virtual tumour when the runs ended. Each dataset contains 100 samples, one for each run. Six virtual tumours produced 12 datasets each and four produced six datasets each, meaning 96 datasets containing 9600 samples were generated in total. Working with these 96 datasets, we resampled each dataset 100 times, taking 10 samples with replacement and averaging them arithmetically to obtain a sample statistic every time, leading to 100 sample statistics *per* dataset. Their mean and 95% confidence intervals were calculated. The 96 confidence intervals are all smaller than 6% of their respective means. We argue that our decision to perform 10 runs *per* configuration was justified.

The first conclusion of this subsection is that a *MYCN*-amplified clone with a higher fractional composition in a tumour does not enjoy a reproductive advantage relative to one with a lower fractional composition in another otherwise identical tumour. The second conclusion is that an ensemble of 10 runs represents the multicellular model with a particular set of parameters adequately.

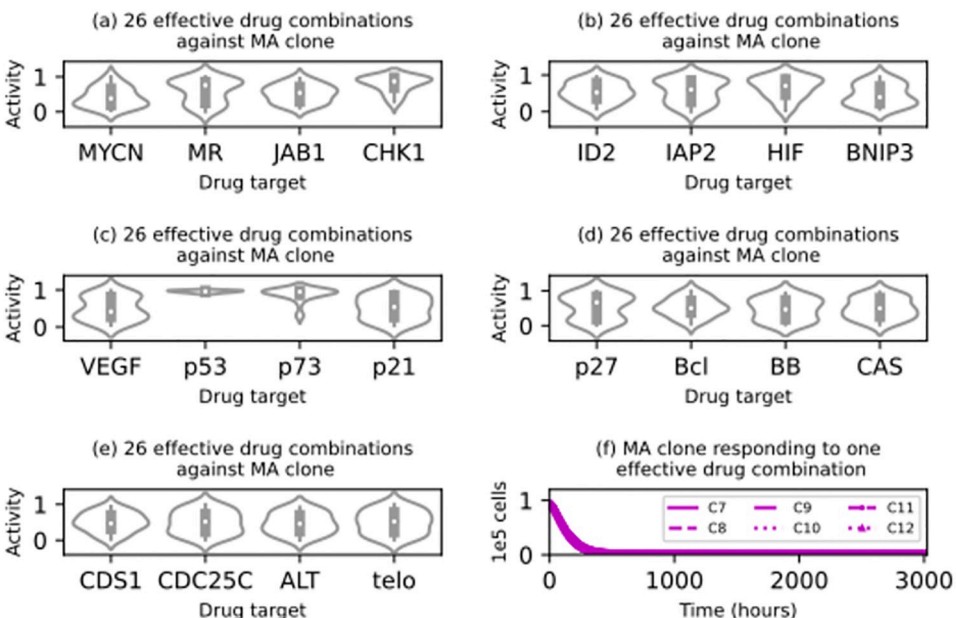

**Fig 10. Results of an *in silico* drug trial for a virtual tumour containing a singular *MYCN*-amplified (MA) clone.** The violin plots from (A) to (E) illustrate how the inhibitory activity levels with respect to each drug target are distributed over the 26 effective drug combinations that led to regression in the corresponding simulations. MR denotes the MAPK/RAS signalling pathway, Bcl denotes Bcl-2 and Bcl-xl collectively, BB denotes Bak and Bax collectively, and telo denotes telomerase. (F) Six time series illustrating how the six subclones within the MA clone responded to an effective drug combination in the corresponding simulation.

## 2.4 Evaluation of combination therapies

Next, we attempted to identify drug combinations that were effective against each clone. The experimental details are provided in subsubsection 5.6.4. An effective combination should lead to a regressed virtual tumour: i.e. no living neuroblastoma cell agents, at the end of its corresponding simulation. In the *in silico* trial targeting a singular *MYCN*-amplified clone, we found 26 such drug combinations. Within this set of drug combinations, their inhibitory effects on the 20 drug targets are distributed as illustrated in the violin plots in Fig 10A–10E. They are marked by their strong inhibitory effects on p53, p73, and CHK1. Fig 10F presents the *MYCN*-amplified clone's response to one of the 26 drug combinations in the corresponding simulation, confirming its effectiveness. The analogous results of the other three trials targeting a singular wild-type clone, *TERT*-rearranged clone, and *ATRX*-inactivated clone respectively are qualitatively similar. Although the six trajectories in Fig 10F look similar, they are actually consistent with our other results. On average, the six subclones within the singular *MYCN*-amplified clone dropped below 100 living neuroblastoma cell agents after 906 (ID: 7), 900 (ID: 8), 919 (ID: 9), 779 (ID: 10), 782 (ID: 11), and 786 (ID: 12) hours. The results presented in subsection 2.1 suggest that inactivating the p53 pathway makes a neuroblastoma cell agent less fit, possibly because the pathway repairs its damaged DNA. A fitter clone or subclone contained more of the living neuroblastoma cell agents in a virtual tumour at the end of the relevant simulation than a less fit one. Consistently, in this simulation, the three *MYCN*-amplified subclones with a wild-type *p53* gene survived more than 100 hours longer than the other three subclones.

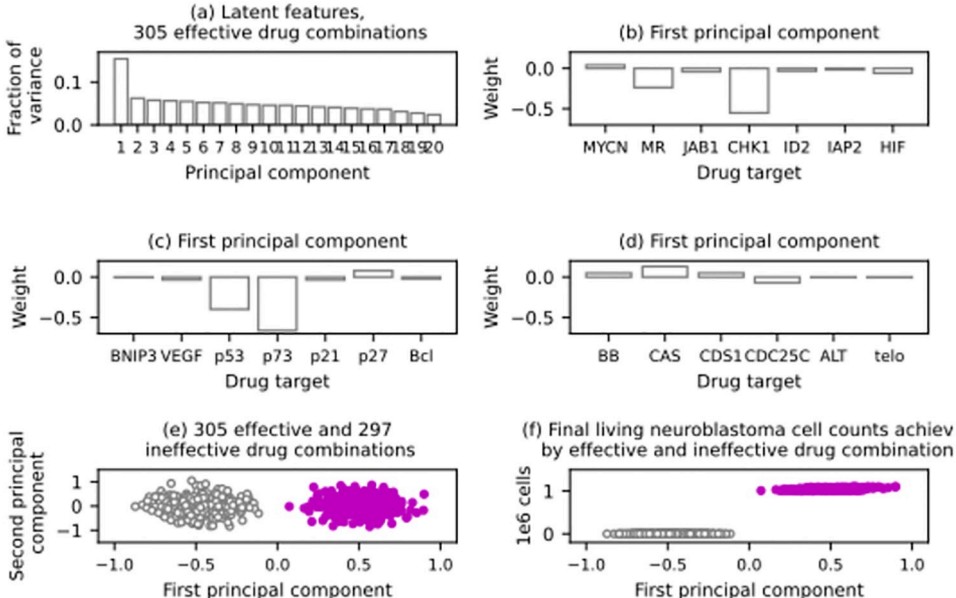

**Fig 11. Results of an *in silico* drug trial for a virtual tumour containing a singular *MYCN*-amplified (MA) clone.**
(A) The outcome of a principal component analysis of a dataset relating to 305 effective and 297 ineffective drug combinations against the virtual tumour; each bar quantifies the amount of variance in the dataset that can be explained by a particular principal component. (B) to (D) The weights of the first principal component of the dataset. Each bar quantifies the importance of inhibiting a drug target, including MYCN; the MAPK/RAS signalling pathway (MR); JAB1; CHK1; ID2; IAP2; HIF; BNIP3; VEGF; p53; p73; p21; p27; Bcl-2 and Bcl-xl (Bcl); Bak and Bax (BB); CAS; CDS1; CDC25C; ALT; and telomerase (telo). (E) The scatter plot projects the entire dataset on its first two principal components and colours two clusters predicted by hierarchical clustering. (F) The scatter plot presents the final outcomes of the simulations associated with the 602 drug combinations in the dataset and colours the same two clusters.

Finally, we used unsupervised machine learning techniques to explore any mechanisms by which the identified drug combinations may shrink each clone. As described in subsection 5.7, we created a dataset comprising the ineffective and effective drug combinations found in each trial. A principal component analysis was performed on each dataset. Fig 11A shows the latent features identified in the dataset relating to the virtual tumour with a singular *MYCN*-amplified clone. The first principal component can explain around 16% of the variance in the merged dataset (602 drug combinations). We performed a principal component analysis on the full set of drug combinations, which are 5000 near-random sets of 20 inhibitory activity levels. As expected, the first principal component of the full dataset can only explain around 5% of the variance therein. On this basis, the first principal component in Fig 11A is likely to be a latent feature indicating a mechanism involving several drug targets. Fig 11B–11D reveals these drug targets. In this drug trial, inhibiting p73, CHK1, p53, and, to a lesser extent, MAPK/RAS signalling was sufficient to shrink the *MYCN*-amplified clone regardless of the other inhibitory activity levels. We validated our prediction by hierarchical clustering, as explained in subsection 5.7. After finding two clusters, we tested their compactness and distinctness by calculating the mean silhouette coefficient over the entire merged dataset, obtaining a score above 0.82, suggesting that the two clusters are compact and well-separated. This can also be seen in Fig 11E, which plots the 602 drug combinations (every example in the merged dataset) on the first two principal components. It colours the two clusters predicted by hierarchical clustering. Finally, the two clusters in Fig 11F match the identified effective and ineffective

drug combinations perfectly. The analogous results of the other three trials are qualitatively similar.

## 3 Discussion

The multicellular model presented in this paper, developed as a part of a multi-scale orchestrated computational framework [1, 2], was intended to capture the cellular dynamics underlying the tumour-level phenomena of neuroblastoma. Simulations based on the multicellular model alone revealed how a *MYCN*-amplified clone is non-linearly related to the other clones (with other mutations) commonly found in a neuroblastoma tumour, its gene expression profile, the tumour's clonal composition, and its microenvironment. These results are consistent with the results of many earlier studies [14, 15, 40, 45–49]. Finally, we identified a promising drug combination targeting p53, p73, and CHK1.

In this section, we will contextualise these results within the literature, first within a biological context and then a technical one. Then, we will return to the study itself to examine our design choices and assess its validity. Finally, we will explore avenues of further research.

### 3.1 The roles of p53 and p73 in neuroblastoma

When we built the multicellular model as a part of the PRIMAGE project, we did not aim to discover any intracellular mechanisms or drug combinations. However, thanks to its versatility, it led to serendipitous discoveries in both areas. We identified *p73* as the most important gene in a *MYCN*-amplified clone (subsection 2.2 and Fig 9). In our simulations, only the *MYCN*-amplified clones that expressed *p73* strongly could expand. We also discovered that inhibiting p53, p73, and CHK1 shrank these *MYCN*-amplified clones. To interpret these results, it is necessary to consider how MYCN, p53, p73, and CHK1 influence each other.

The *p53* gene is rarely mutated at diagnosis [50]. On the other hand, MYCN activates apoptosis-inducing genes when *MYCN* is not amplified, but when *MYCN* is amplified, it activates apoptosis-suppressing genes [18]. For example, as reviewed [47], MDM2 negatively impacts p53 at the protein level (apoptosis-suppressing) in many ways, including degradation and transcriptional inactivation. Although the complex crosstalk between MYCN and the ARF/MDM2/p53 axis [17] within a neuroblastoma cell is not represented explicitly in our model, the negative causal link between MYCN and p53 is captured in a phenomenological manner (subsection B.1 in S1 Text). The downstream effects of p53 are diverse and contradictory, including DNA repair [51], cell cycle arrest by upregulating p21 and p27 [52–54], and apoptosis by upregulating CAS [55, 56]. In response to detecting damaged DNA [54, 57], CHK1 upregulates p73 [54, 57] and downregulates CDC25C to arrest the cell cycle [54]. The downstream effects of p73 [55, 56, 58] are the same as those of p53 except that it does not upregulate p21 and p27. Therefore, p73 can be considered redundant when p53 is functional and active.

Taken together, the simulation results and the literature reviewed by us, collectively indicate that p53 and p73 are more pro-survival than pro-apoptosis in the context captured by our model and its parameters. A corollary is that inhibiting p53, p73, and CHK1 (which activates p73) could promote neuroblastoma regression. On the surface, this conclusion is counter-intuitive. p53 is considered a tumour suppressor and most therapeutic strategies involving it aim to restore its function [47, 48]. For example, the small molecule inhibitors nutilin-3 and MI-219 work by preventing MDM2 from inactivating p53 [59]. If one is prepared to challenge this dogma, however, the idea of inhibiting p53 and p73 to induce neuroblastoma regression is actually worth exploring, perhaps *in vitro* in the first instance.

p53 is a transcription factor that orchestrates the expression of hundreds of target genes, which induce different cellular outcomes in different contexts [46]. There are multiple reasons

for these differences. First, different DNA-damaging stimuli induce qualitatively and quantitatively different dynamics of p53, ranging from pulses to a sustained signal [46]. Second, p53 does not only influence its target genes, but also the dynamics of any connected signalling pathways at the protein level [46]. For example, it was observed in an experiment that although DNA breaks and oxidative stress generated comparable p53 dynamics, they resulted in opposite cellular outcomes [60]. This happened because the dynamics of p53 and MAPKs (ERK, JNK, and p38) interacted to control the balance between cell growth and death [60]. Third, thanks to post-translational modifications and other mechanisms [46], comparable p53 dynamics can be decoded differently because p53 can be modified to allow it access to different promoters to activate different genes.

Of course, the model is sensitive to the assumptions and simplifications made in relation to the intracellular details, but it still represents many aspects of what is known about neuroblastoma cells. In light of the complex manner in which p53 dynamics are encoded and decoded, as well as the generally heterogeneous nature of neuroblastoma [12–15], we hypothesise that there are neuroblastoma cells whose survival depends on p53 activity, possibly because p53 repairs its DNA without inducing apoptosis. This hypothesis rests on the assumption that when a cell's DNA is damaged beyond a threshold, it cannot function as a living system. Another mechanism consistent with our hypothesis is that p53 can transcribe an anti-apoptotic gene, *HO-1*, to promote cellular survival in response to oxidative, nitrosative, hypoxic, and endoplasmic stresses in some cell types [61]. On the basis of this hypothesis, we propose an unorthodox therapeutic strategy targeting p53 and its functionally redundant family member, p73. This alternates between inhibiting MDM2 to restore p53 activity and inhibiting ARF to attenuate p53 activity (by upregulating MDM2) [17, 48]. As a tumour is limited by spatial constraints and the availability of nutrients, this cycling strategy would cause the two populations therein—one suppressed by p53 and one dependent on it—to compete without letting either dominate the tumour, thus preventing overall tumour growth indefinitely. As one population is more resistant to DNA-damaging stimuli and other cellular stresses, the proposed strategy may synergise with traditional chemotherapy and therapies targeting aspects of the tumour microenvironment. Another attraction is that the results of our *in silico* drug trials indicate that this strategy is effective against all four clones.

## 3.2 Towards a digital twin for neuroblastoma research and management

Due to a number of features of the multicellular model, it could fill a notable gap in the field of neuroblastoma modelling, whose main aim is to build a digital twin for neuroblastoma research and management.

The potential of systems modelling to revolutionise neuroblastoma research and treatment was recognised as early as in 2010 [22]. Significant progress has been made at the subcellular level [32–35]. One model describes 98 KEGG pathways, 1287 signalling circuits, and 3057 genes associated with cancer hallmarks [32]. It supports *in silico* trials of targeted therapies acting on these molecular mechanisms. Another model [34] comprises a set of ordinary differential equations that describe the PI3K-AKT, MAPK, mTOR, Ras, and neurotrophin signalling pathways, as well as resistance to EGFR tyrosine kinase inhibitors and an activating mutation in *ALK*. They are represented as a network of 93 species interacting in 85 reactions. Functionally, this network regulates the abundance of PD-L1 in a neuroblastoma cell. The model was used to study how two targeted therapies, gefitinib and crizotinib, modulate the abundance PD-L1 in the cell and hence its susceptibility to immune checkpoint inhibitors.

Before we can use these detailed subcellular models to conduct *in silico* drug trials and personalise therapies, we must couple them to the population dynamics at the whole-tumour

scale, including the effects of evolution, the unique dynamics of small populations, and spatial effects [62, 63]. For example, evolutionary therapy has been demonstrated to be effective in treating breast cancer [64] and castration-resistant prostate cancer [65]. The population-level models recorded in the neuroblastoma modelling literature cannot serve this purpose because they are either too specific to particular drugs or lacking in mechanistic details. For example, a semi-mechanistic model was built to explain the metastatic process in terms of growth and dissemination, but the model does not describe the influence of drugs [66]. At the other end of the spectrum, the influence of bevacizumab, a targeted therapy, on neuroblastoma growth has been modelled in terms of population dynamics [67]. The relationship between neuroblastoma cells and the immune system has also been modelled at the population level within the narrow context of Celyvir, a kind of oncolytic virotherapy [68]. With respect to traditional chemotherapy, the population dynamics of neuroblastoma and immune cells in the presence of topotecan were modelled over a decade ago, but both models neglect evolutionary principles and drug resistance [69, 70]. Our recent paper details the first attempt to leverage evolutionary principles to optimise chemotherapy for neuroblastoma treatment in a patient-specific manner [71]. However, the model captures neither spatial nor stochastic effects, which are important in small populations.

Thanks to its modularity, high resolution, and scalability, our multicellular model can at least partially address these issues. Although the challenges of calibrating it accurately, integrating it with real-time data, and validation remain, we are confident that it is an important piece of jigsaw in the quest for a digital twin of neuroblastoma.

**3.2.1 Modularity.**   Being agent-based, our model enjoys the advantage of modularity, which allows it to incorporate different subcellular models and process their outputs at the population level, thus making it useful for evaluating targeted therapies, combination strategies, and the immune system's effects on neuroblastoma cells. This feature is enhanced by FLAME GPU 2, which supports multi-scale hierarchical simulations. For example, the mechanical model operates at a different time scale from the agents. Additional biological elements could be included similarly by separating time scales.

Targeting specific molecular aberrations is a promising therapeutic strategy [59, 72]. The two elaborate subcellular models reviewed above [32, 34] could be encapsulated within each neuroblastoma cell agent. After integrating non-linear interactions across multiple scales, the hypothetical combined model could be used to assess how targeted therapies such as ALK and MEK inhibitors, alone or in combination, may result in emergent phenomena at the population level. As the use of targeted therapies is an active area of research, a modular model has the extra advantage of being able to accommodate regular updates to the subcellular models.

We agree with the conclusion of a recent review [73] that rational combination strategies are necessary to advance therapies for high-risk neuroblastoma patients by increasing their response rates to first-line therapies, developing effective salvage therapies for relapsed/refractory cases, and keeping them in remission. For example, the ALK inhibitor lorlatinib and the MDM2 inhibitor idasanutlin have been demonstrated to synergise with conventional chemotherapy in preclinical models [74]. As the mammalian cell cycle is coded within each cellular agent, our multicellular model can already be used to evaluate combination therapies involving traditional chemotherapeutic agents, some of which are dependent on the cell cycle [71]. The hypothetical combined model mentioned above could be used to accelerate this line of research by providing mechanistic insights and identifying promising combinations.

Since it is known that high-risk neuroblastomas are able to suppress anti-tumour immunity via multiple mechanisms [73, 75, 76], one promising avenue of research is to develop new immunotherapeutic drugs or combine an existing one with another therapy. For example, combining anti-GD2 antibodies with traditional chemotherapy has shown promise for

relapsed/refractory cases [77]. If the reviewed computational model of PD-L1 expression [34] was incorporated into every neuroblastoma cell agent and new agents were added to represent immune cells, the updated multicellular model could be used to evaluate PD-1 and PD-L1 inhibitors. More generally, it would be a platform to explore novel immunotherapeutic strategies. This idea also aligns with the parallel push for a digital twin of the immune system [78]. There are multi-scale agent-based models of many types of immune cells [79–81], which could be integrated with our model.

**3.2.2 High resolution.**   The ability of our cell-based model to track individual cells means it can relate subtle changes in/among them (internal attributes and cell-cell interactions) to their effects on the whole population. As we demonstrated in a recent study [71], clonal evolution is a process that can be exploited to optimise chemotherapy. The ordinary differential equations we devised can only model the dynamics of nine populations (clones). Thanks to its high resolution, our multicellular model allows a tumour's clonal composition to be defined at the scale of a single cell. If it could be scaled up, one could use it to test the effects of targeting small subclones on the evolutionary dynamics at the whole-tumour level in a patient-specific manner. This hypothetical application might reveal subtle interventions that can drive this tumour to evolutionary dead ends. For example, during our simulations (Fig 8), we observed that within a clone without *MYCN* amplification, its six subclones behaved differently. Specifically, the subclones without inactivating *p53* mutations expanded at the expense of those with these mutations.

**3.2.3 Scalability.**   Our multicellular model's scalability makes it highly versatile. Just by changing the number of cells, the user can already use it to simulate the unique dynamics of small populations. In its current form, it can also mimic a three-dimensional multicellular tumour spheroid. Contingent on future developments, it could become a digital twin of neuroblastoma.

Although it is widely accepted that no drug can directly kill every single cancer cell in a tumour, it is also widely accepted that small populations of cells are more vulnerable to tiny and stochastic perturbations, which are sometimes sufficient to drive a small population to extinction [82]. For example, in a small population, a small increase in the average death rate due to spatial and temporal fluctuations in its surrounding blood flow will have a large effect on the population's long-term prospects. Since our multicellular model treats every cell as a discrete agent defined precisely in space, the model can be configured to study the unique dynamics of small and fragmented populations. For example, the six trajectories in Fig 10F look very similar, but when the cell counts therein are in the hundreds, they reveal small differences in fitness between the six subclones, which could lead to drastically different clinical outcomes. Generalising from this example, we argue that the stochastic nature of our multicellular model allows it to connect minute variations at the cellular level to clinically observable measurements at the tissue level.

Three-dimensional multicellular tumour spheroids play an essential role in modern cancer research due to their close resemblance to *in vivo* solid tumours in some respects [83]. However, several challenges are hindering their adoption in the preclinical phase of drug discovery. For example, mass-producing spheroids with consistent sizes and shapes is not an easy task [83]. At the very least, our model can provide a benchmark against which the robustness of physical spheroids can be assessed. This claim is backed up by the fact that the simulations reported throughout this paper were set up to describe such multicellular structures. More ambitiously, contingent on the developments discussed in subsubsection 3.2.1, it could streamline preclinical drug trials by supporting high-throughput *in silico* experiments to identify promising drug candidates, similar to the simulations reported in subsection 2.4.

Ultimately, the aim is to create a digital twin to enable patient-specific whole-tumour simulations. The main technical bottleneck is that it is not feasible to initialise the model with enough cell agents to describe a whole primary tumour. Surrogate modelling with supervised machine learning methods, such as multiple linear regression and multilayer perceptron models, is a potential remedy. Alternatively, a particularisation and homogenisation model can reduce the computational cost of a simulation by sampling representative regions of the tumour being simulated and interpolating the multicellular simulation results therein to parameterise a population-based model of the whole tumour. In fact, while working with the multi-scale model built for the PRIMAGE project [2], we used this technique to bridge the subcellular, multicellular, and whole-tumour scales by integrating phenomena spanning nine orders of magnitude in space and time [25]. This approach allowed the PRIMAGE consortium to simulate the dynamics of a real patient's primary tumour during induction chemotherapy (80 days).

## 3.3 Design choices and limitations

By definition, a model is a simplified representation of a real system, so design choices must be carefully considered and modellers must be conscious of the implications of these decisions.

**3.3.1 Simplification of intercellular signalling.**   Intercellular signalling (juxtacrine and paracrine) is modelled simplistically with the continuous automaton. A 3D von Neumann neighbourhood update rule reflects the contact-dependent nature of juxtacrine signalling. Paracrine signalling works in a diffusive process. Similar to how oxygen is represented in the multicellular model, a quasi steady state assumption applies to paracrine signalling in its spatial domain. Our collaborator's diffusion model at the whole tumour–level [2] can account for spatial variations among distinct patches of tissues, so we opted to not consider this reactive transport phenomenon (paracrine signalling) explicitly within our model (one patch of tissue). Of course, the assumption that paracrine signalling is an order of magnitude less effective than juxtacrine signalling is naive and *ad hoc*, but without expanding the scope of our collaborator's diffusion model, the precise difference cannot be estimated accurately anyway.

**3.3.2 Inclusion of two types of cellular agents.**   One important choice we made was the decision to include two agent types only: neuroblastoma and Schwann cell agents. Of course, the vasculature and the immune system are also important components of a tumour [75, 76, 84]. However, given that the primary interest of the PRIMAGE project was to optimise chemotherapy [1], we decided to abstract the immune system and represent it as a simple parameter: the probability that an apoptotic or necrotic cell is removed in a single time step. This imitation of how immune cells act against apoptotic and necrotic cells is simplistic and advantageous for model expansion. For example, just considering macrophages alone, there are different modes of recognition for apoptotic and necrotic cells [85]. With its simplistic setup, our multicellular model can be integrated with models of various immune cells [79–81] and immune checkpoint expression [34] to couple extra mechanisms to the existing mechanisms without too many compatibility issues. Our collaborator's diffusion model at the whole tumour–level [2] can account for the condition of a patient's vasculature, so we decided not to model it at the tissue level. We opted for a single parameter: the rate of oxygen supply, which is an output from the diffusion model.

**3.3.3 Computational performance and optimisation.**   We decided to model each cell as an autonomous agent. Despite the computational costs incurred (see below), our decision was justified and remains defensible in hindsight due to the advantages discussed in subsection 3.2. As discussed in subsubsection 3.2.3, the drawbacks can be mitigated by modern computational methods.

On the other hand, the high computational costs necessitated our use of modern GPUs and FLAME GPU 2 to implement the model in the presented simulations. FLAME GPU 2 is designed to enable high-performance GPU-accelerated agent-based simulations. As a result, enhancing computational performance only required limited additional work during the project.

The iterative nature of the cell-cell mechanical model is the most expensive component of the multicellular model. In a test, it initially took 67 iterations to reach a mechanical equilibrium before reducing to a minimum of three iterations at a later agent-level time step. We decided to simplify the mechanical model. For example, we chose not to model cell-cell adhesion. Within FLAME GPU 2, the cell agents in a simulation were sorted according to their spatial coordinates to accelerate the simulation.

As expected, a strong correlation between the initial volume in a simulation and the total runtime was observed in the experiments presented above. As reported in subsection 2.4, four sets of drug trials were conducted. In each trial, each of 5000 combination therapies was assessed in 10 runs lasting 3024 one-hour time steps *per* run. Slightly more than 150 million hours of tumour growth was simulated *per* trial. A trial required between 48 and 50 hours of computational time on four NVIDIA A100 GPUs, provided by the University of Sheffield's High Performance Computing service. At the start of each run, the volume was set to around $8 \times 10^9$ cubic micron (approximately 750 thousand cells). Typically, the simulated cell population would decline due to treatment, but without modern GPUs and FLAME GPU 2, these simulations could not have been performed.

**3.3.4 Intracellular mechanisms.** We designed the internal structures of the two agent types based on the intracellular mechanisms reported in the literature, especially the mechanisms regarding cell cycling, cell death (apoptosis and necrosis), and DNA damage/repair. We did not incorporate the reviewed subcellular models [32, 34] for two reasons. Our collaborators within the PRIMAGE project developed a machine learning–based model to describe the effects of chemotherapeutic drugs on the gene products within our neuroblastoma cell agents [2]. Furthermore, we did not have access to any data to parameterise a subcellular model describing thousands of genes.

Based on the literature, we devised interdependent conditional statements to model each cell agent's internal state transitions, resulting in a network of 20 gene products. One limitation of this static representation is that it neglects the dynamics inside the agent within one time step. For example, it cannot capture feedback control or account for multiple steady states, if they exist. However, the coarse-grained nature of the agent's internal structure would nullify the advantages ordinary differential equations and Boolean networks would offer anyway. As stated in subsection 3.1, p53 alone is capable of activating hundreds of target genes and its dynamics are subject to multiple layers of regulation [46]. Without these intracellular details, the use of a sophisticated subcellular model would have introduced additional parameters without any appreciable improvement to the study. In spite of the simplifications, the multicellular model did help us identify the most sensitive intracellular components in a neuroblastoma cell agent: p53, p73, and CHK1.

## 3.4 Validity

The simulation results presented in this paper lend credence to the multicellular model's validity. We successfully produced the intuitively correct phenomenon of hypoxia-driven regression [40]. Consistent with the literature [14], a *MYCN*-amplified clone's expansion in a heterogeneous virtual tumour was contingent on the latter's conditions, especially its genetic profile. According to the cited paper [14], while numerical chromosomal aberrations favour

such a clone, segmental chromosomal aberrations constrain it. Our results indicate that gene expression is another potential determinant. Again consistent with the literature [15] is that a more dominant *MYCN*-amplified clone did not enjoy a significant reproductive advantage over its peers in our simulations. Despite the incomplete and simplistic representation of the known intracellular mechanisms in the model, it still helped us identify p53, p73, and CHK1 (a proxy for p73) as the most important gene products in a neuroblastoma cell. The importance of p53 is indisputable based on the evidence in the literature [45–48]. Not only is *p73* rarely lost or mutated in cancers, but it is also often over-expressed in cancers and it can compensate for *p53* [49], just as our simulation results suggest.

Despite the *ad hoc* choice of 10 repeats *per* configuration/simulation, the average results are robust as indicated by the confidence intervals we built by resampling them. On this basis, we argue that the computational costs associated with performing more than 10 runs *per* configuration would not have been worthwhile.

The biggest cause for skepticism comes from the parameters. We encountered the common problem of lacking suitable human-derived data for calibration and validation. We tackled this issue by using mostly *in vitro* experimental data, which may not be valid in *in vivo* scenarios. As discussed in our other paper [2], phenotypic datasets at the cellular level were simply not accessible through *in vivo* imaging or other spatially aggregated modalities.

### 3.5 Future work

Now that we have built and partially validated the multicellular model, one avenue of further research is to make the intracellular details more realistic by incorporating the reviewed subcellular models [32, 34] to regulate cell cycling, cell death, and DNA damage/repair. Armed with a more sophisticated model (notwithstanding the challenges relating to parameterisation, as discussed above), we will revisit our hypothesis that a high-risk tumour contains two distinct populations of neuroblastoma cells wherein p53 promotes cell survival and cell death respectively. The ability to predict the abundance of PD-L1 in each neuroblastoma cell agent [34] will pave the way for additional agent types to be added to represent the immune system, thus enabling us to simulate immunotherapy. Addition of new agent types, especially one representative of endothelial cells, will allow the tumour vasculature to be modelled directly. Coupled with a set of reaction-diffusion-advection equations, another useful addition, this virtual vasculature will help the model capture more precise spatial effects. In parallel to model development, we will further explore surrogate modelling with supervised machine learning as a means to save computational costs. Most importantly, we recommend that an *in vivo* experimental study dedicated to measuring the model parameters be conducted.

### 4 Conclusion

During the PRIMAGE project, we developed—to the best of our knowledge—the first multicellular computational model of neuroblastoma. As reported in our recent paper [2], we implemented and partially validated it before integrating it into a multi-scale orchestrated computational framework. This follow-up paper complements the earlier publication, by reporting what was accomplished with the multicellular computational model alone. By implementing it on GPUs and performing large-scale dynamic simulations, we made discoveries regarding the nature of the disease and potential therapeutic strategies. The results, when interpreted in the context of what is known about neuroblastoma, suggest that the role of p53 and its functionally redundant family member, p73, is not clear-cut. Our hypothesis is that p53 serves contradictory purposes in two distinct neuroblastoma cell populations in a tumour. In one population, it is pro-survival. In the other, it is pro-apoptosis. Consistent with this

hypothesis is an unorthodox therapeutic strategy. In this hypothetical scenario, an oncologist would alternate between inhibiting MDM2 to restore p53 activity and inhibiting ARF to attenuate p53 activity (by upregulating MDM2). Due to the multicellular model's modularity, it can easily incorporate multiple subcellular models and additional cellular agents, thus allowing targeted therapies, combination therapies, and immunotherapies to be simulated. Its cellular resolution means it can be used to simulate tumour evolutionary dynamics in detail. Due to its scalability, it has a wide range of applications, including the potential to become the foundation of a digital twin of neuroblastoma.

## 5 Materials and methods

This section gives an overview of the model components, the stochastic simulation algorithm, the calibration procedures, how the simulations were configured, and the computational tools used to produce and analyse our simulation results. S1–S4 Text files provide full details relating to the model and the algorithm, including the behavioural rules applied to the cell agents.

The multicellular model is comprised of a continuous automaton describing the tumour's microenvironment, autonomous discrete agents representing neuroblastoma and Schwann cells, and a centre-based mechanical model of cell-cell repulsion. The continuous automaton represents the tumour microenvironment as a grid-like structure wherein each voxel is associated with continuous variables such as the oxygen level therein. Each discrete cell agent is defined by various attributes, including its cell cycle position, mutations, gene expression pattern, and more. Its behaviour is stochastically dependent on these attributes and includes cell cycling, cell death, and more. The centre-based mechanical model describes the properties of these agents as physical objects, describing how they repel each other as soft spheres. The simulation algorithm integrates these model components by orchestrating information flow between them, repeating a finite set of instructions (including stochastic cellular decisions) in a fixed order. The model was calibrated using mostly *in vitro* data and some of its parameters were refined for *in vivo* scenarios. Simulations were then configured to understand clonal competition in heterogeneous tumours, assess the *MYCN*-amplified clone's sensitivity to various factors, and conduct an *in silico* trial of 5000 hypothetical drug combinations.

### 5.1 Continuous automaton

Mathematically and geometrically, this is a grid of voxels representing the model's spatial domain. Biologically, it describes the tumour's microenvironment in terms of the spatial distributions of cells and extracellular matrix therein. Each voxel is centred around a Cartesian coordinate and spans 30 microns in each spatial dimension. There is an 11-dimensional state vector ($voxstate_{i,j,k}$) associated with it, indicating the numbers of neuroblastoma cells (total), apoptotic neuroblastoma cells, necrotic neuroblastoma cells, living neuroblastoma cells, Schwann cells (total), apoptotic Schwann cells, necrotic Schwann cells, living Schwann cells, and matrix-producing Schwann cells (a subset of living Schwann cells); the total number of cells; and the fraction of volume occupied by extracellular matrix therein.

A voxel's 3D von Neumann neighbourhood includes the voxel itself and its six orthogonally adjacent voxels. The abstracted effects of juxtacrine (contact-dependent) and paracrine (diffusive) signalling on an agent in the voxel depend on the states of the voxels within its 3D von Neumann neighbourhood. As juxtacrine signalling depends on direct cell-cell contact, the signalling molecule (ligand) does not travel through an extracellular space and has less time to degrade, so paracrine (diffusive) signalling is assumed to be an order of magnitude (precise value chosen *ad hoc*) less effective than juxtacrine (contact-dependent) signalling.

Several variables describing the tumour's microenvironment are not spatially resolved, but for the sake of simplicity, they notionally belong to this continuous automaton. They are the oxygen level, which is assumed to be uniform (if $static_{O2} == 1$, as explained in S1 Text), the rate of oxygen supply, which is also assumed to be uniform (if $static_{O2} == 1$, as explained in S1 Text), the progress of angiogenesis, and the number of VEGF-producing (angiogenesis-promoting) neuroblastoma cells. Since the diffusion distance of oxygen in tumour tissues typically ranges from 100 to 150 μm [86] and our spatial domain is a cube spanning 2 mm (2000 μm) in each dimension, this is an idealised representation of the vasculature. The assumption is that the spatial domain has a uniform and well-perfused vasculature when $static_{O2} == 1$. When $static_{O2} == 0$, two further assumptions are that the angiogenesis-promoting neuroblastoma cells collectively affect every part of the vasculature to the same extent and that each of them can access the total oxygen supply from the vasculature to the same extent. These assumptions are of course incorrect, due to the known fact that tumour vasculatures are usually chaotic [87, 88]. The missing details are supplied by another component within the multi-scale orchestrated computational framework encompassing the multicellular model [2], wherein vascularisation and cellularity data extracted from dynamic contrast-enhanced magnetic resonance imaging (DCE-MRI) and diffusion-weighted imaging (DWI) are applied to the whole tumour–level transport model, so tumour heterogeneity is already handled at a higher spatial scale.

## 5.2 Autonomous discrete agents

Neuroblastoma and Schwann cells, the two main cell populations in a neuroblastic tumour [38], are modelled as autonomous discrete agents (soft spheres) which populate the CA grid.

Each neuroblastoma cell agent's state is defined by four types of attributes, which are recorded in four vectors. The physical vector ($physical_n$) records the cell's spatial coordinates, the net force acting on it, its total overlap with the other cells in the system, the number of cells within its search distance, and whether it is mobile. More details on these physical aspects will be provided in the next subsection. The cellular vector ($cellular_n$) contains information about its position in the cell cycle, how differentiated it is along the axis from a neural crest cell to a healthy neuron, its susceptibility to cell death (apoptosis and necrosis), the state of its telomeres (length), the state of its DNA (normal, damaged, or unreplicated), and its metabolic state (whether it is hypoxic and whether it has enough ATP). The mutation vector ($mutation_n$) indicates whether its *MYCN* gene is amplified, whether its *TERT* gene is rearranged, whether its *ATRX* gene is inactivated, and the status of its *ALK* gene. The molecular vector ($molecular_n$) indicates whether the cell has sufficient ATP, whether its telomerase and alternative lengthening of telomeres (ALT) mechanism are active, and whether its other 18 sets of gene products are active.

Although in theory, the mutation and molecular vectors of a neuroblastoma cell agent can be given any values, only a limited set of combinations were used in our study. These restrictions reflect the population structure summarised in Fig 1E. Each neuroblastoma cell agent belongs to one clone and one subclone. The consensus is that *MYCN* amplification, *TERT* rearrangement, and *ATRX* inactivation are mutually exclusive mutations in a neuroblastoma cell [19, 20], so the population structure comprises four clones: one for each mutation and one without any of the three (the wild type). Each clone is further divided into six subclones defined by whether and which parts of the MAPK/RAS and p53 pathways are mutated [20]. In the molecular vector, each of the two pathways is represented as a Boolean variable. The probability that each pathway is active depends on whether and how it is mutated. The mutations inactivating the p53 pathway are lumped into one mutation in our model; it switches off the

corresponding Boolean variable. With respect to the MAPK/RAS pathway, *ALK* activation and its amplification are collectively represented as one mutation, while any other activating mutations are combined into another mutation. With these mutations, there is a higher probability that the MAPK/RAS pathway is active. Less intuitively, *MYCN* amplification is also associated with the pathway's activation [89]. There is another activating mechanism in reverse. ALK (not the other members in the MAPK/RAS pathway) can also regulate MYCN activity positively [90]. The molecular vector also contains Boolean variables representing whether telomerase, ALT, and 16 other sets of gene products are active. Collectively, its 20 Boolean variables regulate cell cycling, cell death, and DNA/telomere repair in a microenvironment-dependent manner. These regulatory updates are described in subsection 5.4, which presents the stochastic simulation algorithm. The full description of how the gene products regulate each other and cellular phenomena (such as cell cycling and apoptosis) can be found in S1 Text.

Each Schwann cell agent's state is represented by two vectors and an additional Boolean variable. Its physical vector ($physical_{sc}$) is analogous to that of a neuroblastoma cell agent. Its cellular vector ($cellular_{sc}$) is also analogous to that of a neuroblastoma cell agent, except that it does not describe degree of differentiation given that Schwann cells are assumed to be fully differentiated. The additional Boolean variable ($ATP_{sc}$) indicates if the cell has enough ATP. How its state is updated in a microenvironment-dependent manner is explained in subsection 5.4.

During a simulation, each agent integrates signals from its neighbours and microenvironment to update its attributes, including the state of its DNA, the state of each intracellular species, its degree of differentiation, its apoptotic status, and its necrotic status. Then, it attempts to progress in the cell cycle. Each agent's radius increases as it progresses through the G1 and G2 phases of the cell cycle [54]. If it is apoptotic or necrotic, it is stochastically removed from the system with a probability ($P_{lysis}$) of 0.35 [91]), meaning that the space previously occupied by the dead cell agent will become available to its neighbours. Otherwise, it may divide if it is at the end of the cell cycle. S1 Text provides full details relating to these behavioural rules.

## 5.3 Mechanical model

The centre-based mechanical model is an off-lattice model allowing for continuous changes in the spatial coordinates of every agent. It comprises a linear force law and an equation of motion.

The force law relates the overlap between any two cells to the repulsive force acting between them:

$$\delta_{1,2} = R_1 + R_2 - \| \boldsymbol{r_1} - \boldsymbol{r_2} \| . \tag{1}$$

In this equation, $\delta_{1,2}$ quantifies the overlap, $R_1$ and $R_2$ are the cells' radii, which increase throughout the cell cycle, and $\boldsymbol{r_1}$ and $\boldsymbol{r_2}$ are their displacement vectors. These variables are all in microns (μm).

The magnitude of the repulsive force between this pair of cells ($F_{1,2}$ in N) is given by another equation:

$$F_{1,2} = k_1 \delta_{1,2}, \tag{2}$$

where $k_1$ (N m$^{-1}$) is just a linear force law parameter.

Applying Eq 2 to a particular cell and every other neighbouring cell will result in a set of force vectors, the sum of which will give a vector describing the net force on this particular cell. If the cell is mobile and there are enough cells within its search distance to inhibit its growth by contact, this vector will be multiplied by a factor of $k_2$, which models the cell's

intrinsic ability to migrate. Either way, the final net force acting on the cell is denoted by the vector $(F^x, F^y, F^z)$, whose unit is N.

The cell's displacement in each dimension, measured in μm, caused by the corresponding net force component, is given by an equation of motion resolved along each Cartesian axis. For example, the equation relating to the $x$-axis is as follows:

$$F^x = \mu(1 + M)\frac{dx}{dt}, \tag{3}$$

where $\mu$ (N s m$^{-1}$) denotes the viscosity in a voxel without extracellular matrix and $M$ (dimensionless) denotes the fraction of volume occupied by matrix in the voxel wherein the cell resides at time $t$ (h). This fraction, $M$, is a component of the voxel's eight-dimensional state vector (subsection 5.1).

Additional parameters were selected to ensure the mechanical model describes realistic behaviour. For example, the number of neighbouring agents allowed within an agent's vicinity —or rather the radius defining its vicinity—was set to prevent unrealistic tumour growth. Empirical tests were conducted. The maximum displacement experienced by an agent in one force resolution step (36 seconds based on [92]) was found to be less than 0.55 microns in all experiments. The maximum displacement in one agent time step (one hour) comprised of multiple force resolution steps was found to be less than 11 microns in all experiments. The velocity of mesenchymal cell migration in 3D collagen matrices is approximately between 0.1 and 0.5 microns in a minute [93, 94]. Therefore, the observed displacements were judged to be physically realistic and no further refinement was made to the relevant parameter.

## 5.4 Stochastic simulation algorithm

We devised a Monte Carlo algorithm to implement the multicellular model dynamically (source code: link). After a simulation is configured as described below, the algorithm is stochastically initialised. Then, it iterates a pre-defined sequence of operations a finite number of times. Each configuration requires multiple runs of the algorithm. Collectively, the results generated from these runs form an ensemble describing the configuration.

**5.4.1 Initialisation.**   A simulation requires the following inputs, which will be described in detail in the rest of this subsection.

1. The simulation's time scale and the spatial domain's size and cellularity.

2. The split between neuroblastoma cell agents and Schwann cell agents and the neuroblastoma cell agents' degree of differentiation.

3. The neuroblastoma cell agents' mutation profiles, as well as the states of their ALT and p53.

4. Oxygen level in the tumour microenvironment and the chemotherapy regimen.

The time scale is defined by its extent (number of time steps) and grain (duration of each time step), while the spatial domain is a cube defined by its initial volume and cellularity, which is the fraction of the cube accessible to a cell agent. For example, a cellularity of 0.8 means that 80% of it is accessible to the cell agents therein, while 20% is full of extracellular matrix.

The split between neuroblastoma cell agents and Schwann cell agents in the cellular fraction and the neuroblastoma cell agents' degree of differentiation are initialised by specifying the tumour's histological category (neuroblastoma, ganglioneuroblastoma, nodular ganglioneuroblastoma, intermixed ganglioneuroblastoma, ganglioneuroma, maturing ganglioneuroma, and mature ganglioneuroma [95]) and grade of differentiation (undifferentiated, poorly

differentiated, and differentiating [95]). Generally, neuroblastoma is characterised by a smaller fraction of Schwann cell agents and less differentiated neuroblastoma cell agents than ganglioneuroblastoma; ganglioneuroblastoma has the same relationship with ganglioneuroma. By definition, an undifferentiated neuroblastoma cell agent's degree of differentiation is zero. Again by definition, a neuroblastoma cell agent in a mature ganglioneuroma is mature, so its degree of differentiation is 100%. Since we could not find further information regarding this variable, we decided to set up evenly spaced boundaries between the two extreme values. Below, we will explain how we estimated the presented ranges of Schwannian fractions based on the cited study [95].

1. According to [95], Schwannian stroma occupies up to 50% of the tumour tissue in a neuroblastoma. Evenly spaced boundaries were chosen due to the lack of quantitative evidence regarding how undifferentiated, poorly differentiated, and differentiating neuroblastomas differ in this respect. On the other hand, a minimum value of five percent ensures that a system always has some Schwann cell agents. If the histological category is neuroblastoma (0), five to 17% of the cell agents in an undifferentiated tumour are Schwann cell agents, 17 to 33% in a poorly differentiated tumour are Schwann cell agents, and 33 to 50% in a differentiating tumour are Schwann cell agents. In an undifferentiated tumour, $deg_{diff,n}$ is zero in each neuroblastoma cell agent. This variable represents the degree to which the agent is differentiated along the neural crest–derived sympathoadrenal lineage [96] and a tumour comprising more differentiated neuroblastoma cells generally behaves less aggressively [97]. In a poorly differentiated tumour, it is between zero and 20%. In a differentiating tumour, it is between 20 and 40%. The precise value in each range is determined stochastically by the program according to the continuous uniform distribution.

2. If the histological category is ganglioneuroblastoma (1), the more precise category is equally likely to be nodular ganglioneuroblastoma (2) or intermixed ganglioneuroblastoma (3) since there is no evidence favouring one possibility over the other.

3. If the histological category is nodular ganglioneuroblastoma (2), the more precise category is equally likely to be neuroblastoma (0), intermixed ganglioneuroblastoma (3), or maturing ganglioneuroma (5) since there is no evidence favouring one possibility over the other two. It is known that a tumour belonging to this category comprises macroscopic neuroblastic nodules embedded within a tumour tissue resembling intermixed ganglioneuroblastoma or ganglioneuroma [95]. Maturing ganglioneuroma is assumed to be more relevant than mature ganglioneuroma because of its proximity to nodular ganglioneuroblastoma on the differentiation scale.

4. According to [95], Schwannian stroma occupies more than 50% of the tumour tissue in an intermixed ganglioneuroblastoma, which is described as being 'rich' in Schwannian stroma. A tumour belonging to the category of ganglioneuroma is described as having more Schwannian stroma than an intermixed ganglioneuroblastoma and it is described as being 'dominant' in Schwannian stroma. The model assumes that the boundary between 'rich' and 'dominant' is 67%. If the histological category is intermixed ganglioneuroblastoma (3), 50 to 67% of the cell agents in a tumour are Schwann cell agents. In a neuroblastoma cell agent, $deg_{diff,n}$ is between 40 and 60%. This variable represents the degree to which the agent is differentiated along the neural crest–derived sympathoadrenal lineage [96] and a tumour comprising more differentiated neuroblastoma cells generally behaves less aggressively [97]. The precise value in each range is determined stochastically by the program according to the continuous uniform distribution.

5. If the histological category is ganglioneuroma (4), the more precise category is equally likely to be maturing ganglioneuroma (5) or mature ganglioneuroma (6) since there is no evidence favouring one possibility over the other.

6. The boundary separating maturing ganglioneuroma and mature ganglioneuroma lies in the middle of the range (67 to 100%) due to the lack of relevant quantitative datasets. If the histological category is maturing ganglioneuroma (5), 67 to 83% of the cell agents in a tumour are Schwann cell agents. In a neuroblastoma cell agent, $deg_{diff,n}$ is between 60 and 80%. This variable represents the degree to which the agent is differentiated along the neural crest–derived sympathoadrenal lineage [96] and a tumour comprising more differentiated neuroblastoma cells generally behaves less aggressively [97]. The precise value in each range is determined stochastically by the program according to the continuous uniform distribution.

7. The maximum value is 95% and this imposed ceiling ensures that a system always has some neuroblastoma cell agents. If the histological category is mature ganglioneuroma (6), 83 to 95% of the cell agents in a tumour are Schwann cell agents. In a neuroblastoma cell agent, $deg_{diff,n}$ is between 80 and 100%. This variable represents the degree to which the agent is differentiated along the neural crest–derived sympathoadrenal lineage [96] and a tumour comprising more differentiated neuroblastoma cells generally behaves less aggressively [97]. The precise value in each range is determined stochastically by the program according to the continuous uniform distribution.

8. If the initial conditions of the two variables were available, they could be set directly and precisely, and the above estimates would not be necessary.

The agents' state vectors must be configured. With respect to a neuroblastoma cell agent, the mutation vector and two components in the molecular vector (whether ALT and p53 are active) require configuration. By contrast, a Schwann cell agent simply adopts its default attributes.

Two microenvironmental conditions require configuration: the initial oxygen level and the chemotherapy regimen. In this model, chemotherapy inhibits six gene products in a neuroblastoma cell agent: CHK1, JAB1, HIF, MYCN, telomerase, and p53. It is also necessary to specify when chemotherapy is active and the probability with which it inhibits each gene product when active.

After configuration, random spatial coordinates are assigned to each agent and the centre-based mechanical model is solved numerically to minimise the total overlap in the spatial domain. The continuous automaton is set up based on the equilibrated spatial coordinates. The oxygen supply rate therein is initialised based on the assumption that it is equal to the two cell populations' initial consumption rate.

**5.4.2 Simulation.**   Following this initialisation routine, the sequence of operations illustrated in Fig 2 is implemented once *per* time step until the end point is reached.

1. The neuroblastoma cell agents in this spatial domain are evaluated serially. Each agent senses its neighbourhood to determine whether it is hypoxic, nourished, and/or affected by chemotherapy, as well as to integrate the stimuli from the other agents in its neighbourhood. In response, it updates the status of its DNA; its degree of differentiation; whether it is living, apoptotic, or necrotic; and whether its gene products are active. These updated attributes decide whether the agent can progress in the cell cycle. If it is apoptotic or necrotic, it may be removed in an operation representing the immune system's actions. Otherwise, it may divide into two daughter cells.

2. The Schwann cell agents are evaluated in the same way as the neuroblastoma cell agents, with two exceptions. First, the model neither describes their gene products nor links them to their cellular functions. Second, Schwann cells are assumed to be fully differentiated, so the degree of differentiation is not an agent attribute either.

3. The total cell-cell overlap in the spatial domain is minimised using the centre-based mechanical model. The continuous automaton is updated to account for the agents' new spatial coordinates.

4. The oxygen supply rate and oxygen level in the continuous automaton are updated if $static_{O2} == 0$, as explained in S1 Text.

5. The fraction of volume occupied by extracellular matrix in each voxel is updated.

The entire algorithm is repeated multiple times with the same configuration (parameters and initial conditions or the distributions of these quantities) to generate an ensemble of results for the configuration. Full details of the algorithm, including the rules/Bernoulli trials applied to cell agents, are provided in S1–S4 Text files.

## Model calibration

Literature values were initially obtained for some of our model parameters, as indicated by the sources cited in Tables 2, 3, 4, 5, 6 and 7. As explained in subsection 5.3, the mechanical model was calibrated by imposing physically realistic constraints on its parameters. After these procedures, 22 unconstrained parameters related to the following agent attributes and phenomena were identified:

1. In the model, a gene's expression level is represented as the probability that its product is produced when the necessary conditions are all satisfied. The expression levels of *MYCN*, the genes encoding the MAPK/RAS signalling pathway, *p53*, *p73*, and *HIF* in neuroblastoma cell agents required calibration.

2. The two basal cell cycling rates (neuroblastoma cells and Schwann cells), as well as the cellular responses to hypoxia, chemotherapy, impaired DNA, apoptotic signals, and necrotic signals also required calibration.

3. Literature values could not be found for the non-mechanical interactions between the two cell populations ($R^{jux}_{pro,sc}$, $R^{jux}_{diff,nb}$, $R^{jux}_{apop,nb}$, $R^{para}_{pro,sc}$, $R^{para}_{diff,nb}$, $R^{para}_{apop,nb}$, $R_{diff}$) and the removal of apoptotic and necrotic cells ($P_{lysis}$).

We did not actively select these 22 normalised parameters (since values were not available in the literature): rather, values used in later simulations emerged during the calibration process. Since they were entirely unconstrained at the start of the calibration stage, each was sampled from the entire range between zero and one. In the first instance, we calibrated these unconstrained parameters without considering the exact effects of chemotherapy on the gene products represented in our model. Instead, chemotherapy induced apoptosis directly in this set of simulations. After aggregating experimental and clinical data [20, 39–42], we devised a tournament-style calibration pipeline, summarised in Fig 3 and Table 1. We used the Latin hypercube sampling technique to generate 3000 near-random points in the unconstrained parametric space [98]. As described in detail below, we set up six rounds of elimination wherein our aim was to reproduce the aforementioned datasets with the 3000 near-random parametric combinations, eliminating infeasible combinations, until only one remained. On

account of the possibility that the outcome would depend on the order in which the elimination rounds were arranged, we designed the tournament to prioritise *in vivo* data. The rounds were arranged in an increasing order of sophistication, with the first dataset—*in vitro* and without genomic data—being the coarsest and the sixth—clinical, patient-specific, and with known mutations—being the most precise. In other words, out of the aggregated datasets, the more general ones were used for shortlisting rather than fine-tuning.

In each round, we used the combinations of unconstrained parameters from the previous round to attempt to reproduce a specific dataset from the literature. In the first round, we used each of the 3000 combinations to reproduce the growth kinetics of neuroblastoma observed in an *in vitro* study [39]. As reported in chart 1 of this paper, the number of viable cells in their culture was recorded at the start of the experiment, and after 24, 48, 72, 96, and 168 hours. In the first calibration study, the ratio between the number of living cell agents (neuroblastoma and Schwann cells) and the initial number of living cell agents was recorded at the beginning, and after 24, 48, 72, 96, and 168 time steps. For each parametric combination, the six pairs of simulated and experimental cell counts were squared independently and then summed to give the metric of this study. The 1000 combinations—this number was picked informally based on the first round's computational time—that gave the smallest residual sums of squared differences entered the second round. In the second round, each remaining combination's ability to reproduce neuroblastoma's hypoxic response *in vitro* [40] was assessed. 50 combinations entered the third round. Here, they were used to simulate the regulatory dynamics between neuroblastoma and Schwann cells, specifically the regulation of cell cycling and apoptosis. The results were compared to the experimental data from an *in vitro* study [41] to select the top 10 combinations. Knowing from a clinical study that whether a tumour progresses, regresses, or differentiates is dependent on its histological category [42], we used the remaining 10 combinations to estimate the probability of each outcome conditional on the histological category. Table V of the cited paper [42] reports the clinical outcomes of patients whose histological categories were intermixed ganglioneuroblastoma or maturing ganglioneuroma. In our simulations, we mimicked a histological category by initialising the split between neuroblastoma cell and Schwann cell agents, as well as the neuroblastoma cell agents' degree of differentiation. If a run finished without any living neuroblastoma cells, it was considered a regressed tumour. If a run finished with just living neuroblastoma cells that were at least 90% differentiated on average, it was considered a differentiated tumour. If a run finished with just living neuroblastoma cells that were less than 90% differentiated on average, it was considered a progressing tumour. Only cell counts, not tumour volumes, were considered in this classification step. In table V of the cited paper [42], the five-year event-free survival rate of the intermixed ganglioneuroblastoma patients is reported to be 94.1%. Six of the 10 parametric combinations led to progression in at least 64% of the runs simulating intermixed ganglioneuroblastoma patients, so they were eliminated. The remaining four led to regression or differentiation in 100% of the runs. The five-year event-free survival rate of the maturing ganglioneuroma patients is reported to be 100%. The remaining four parametric combinations gave us simulation results also in agreement with this clinical observation. Since they predicted realistic outcomes, they were retained for the next round. In the fifth round, these four combinations were used to reproduce the clinical outcomes of 10 patient groups with different mutations [20], thus eliminating another combination. Supplementary figures S16A and S16B of the cited paper report the clinical outcomes of 43 patients. These were combined to form 10 patient groups based on their mutation profiles and treatment approach. After applying a group's mutation profile to the initial neuroblastoma cell agents in the multicellular model, we simulated its dynamics. Similar to the fourth round, we classified the outcome of a run as regression, differentiation, or progression. One parametric combination consistently failed to produce a progressing virtual tumour to

**Table 2. Parameters constraining the spatially homogeneous variables.**

| Parameter | Meaning | Value | Units | Source |
|---|---|---|---|---|
| $C_{O2}^{s}$ | Concentration scale of oxygen, assumed to be the oxygen concentration in the kidney | 72 | mmHg | [104] |
| $R_{O2}^{0}$ | Oxygen production rate *per* cell | $-1.875e^{-13}$ | moles h$^{-1}$ | [105] |
| $T_{ang}^{c}$ | Time scale of angiogenesis | 100 | h | [106] |
| $P_{necro}^{is}$ | Probability that a necrotic cell triggers a necrotic signal in its 3D von Neumann neighbourhood in an hour | 0.5768 | None | Calibrated |
| $P_{lysis}$ | Probability that an apoptotic or necrotic cell is removed from a simulation, assumed to be the probability that it is engulfed or lysed by an immune cell in an hour *in vivo* | 0.35 | None | [91] |
| $\rho$ | Cell density in space not occupied by extracellular matrix | $9.39e^{-5}$ | μm$^{-3}$ | [107] |

mirror the clinical observations, so it was eliminated. In the final round, we used a different dataset from the same study [20]—*MYCN*-amplified tumours only—to identify the best parametric combination. Supplementary figure S18A of the cited paper reports the clinical outcomes of 49 patients with *MYCN*-amplified tumours. We combined them to form four patient groups based on their mutation profiles. One of the three remaining parametric combinations gave us more realistic simulation results (around half of the runs ended in progression as expected) than the other two when *p53* inactivation was not applied to the model, so it was chosen as the tournament winner.

After the tournament, to render the model suitable for simulating *in vivo* and clinical scenarios, selected model parameters were fine-tuned to account for additional information and data; they were provided by our PRIMAGE partners [1] or found in the literature [99–103]. Then, we set constraints on the entire population's doubling time, the neuroblastoma cell agents' growth and differentiation rates, and the Schwann cell agents' growth rate, thus refining the parameters controlling cell cycling and the abstract representation of juxtacrine (contact-dependent) and paracrine (diffusive) signalling.

The parameters constraining the model's spatially homogeneous variables can be found in Table 2. Those defining the continuous automaton are presented in Table 3. Those shared by both populations of cellular agents are in Table 4. The additional parameters necessary for defining neuroblastoma cell agents and Schwann cell agents are presented in Tables 5 and 6 respectively. Finally, the mechanical model's parameters are presented in Table 7.

## 5.6 *In silico* experiments

We performed four sets of computer simulations to study clonal competition, *MYCN*-amplified clones with different gene expression patterns, how a tumour's macroscopic properties influence its *MYCN*-amplified clone, and combination therapies respectively.

**Table 3. Parameters defining the continuous automaton.** Juxtacrine signalling depends on direct cell-cell contact, so the signalling molecule (ligand) does not travel through an extracellular space and has less time to degrade. As a result, the strength of each paracrine (diffusive) signalling event is assumed to be an order of magnitude (precise value chosen *ad hoc*) smaller than its juxtacrine (contact-dependent) counterpart.

| Parameter | Meaning | Value | Units | Source |
|---|---|---|---|---|
| $L_{voxel}$ | Voxel's side length, assumed to be bigger than a cell's diameter | 30 | *μm* | Assumed |
| $R_{pro,sc}^{jux}$ | Strength of neuroblastoma cells' influence on Schwann cells' proliferation through juxtacrine signalling | $3.74e^{-3}$ | None | Calibrated |
| $R_{diff,nb}^{jux}$ | Strength of Schwann cells' influence on neuroblastoma cells' differentiation through juxtacrine signalling | $5.21e^{-4}$ | None | Calibrated |
| $R_{apop,nb}^{jux}$ | Strength of Schwann cells' influence on neuroblastoma cells' apoptosis through juxtacrine signalling | $3.04e^{-2}$ | None | Calibrated |
| $R_{pro,sc}^{para}$ | Strength of neuroblastoma cells' influence on Schwann cells' proliferation through paracrine signalling | $3.74e^{-4}$ | None | Assumed |
| $R_{diff,nb}^{para}$ | Strength of Schwann cells' influence on neuroblastoma cells' differentiation through paracrine signalling | $5.21e^{-5}$ | None | Assumed |
| $R_{apop,nb}^{para}$ | Strength of Schwann cells' influence on neuroblastoma cells' apoptosis through paracrine signalling | $3.04e^{-3}$ | None | Assumed |

**Table 4. Parameters defining both types of cellular agents.**

| Parameter | Meaning | Value | Units | Source |
|---|---|---|---|---|
| $L_{cell}$ | Diameter of a cell at the start of the cell cycle | 11 | $\mu m$ | [107] |
| $T_{G1}$ | Duration of the G1 stage of the cell cycle | 12 | h | [54] |
| $T_S$ | Duration of the S stage of the cell cycle | 6 | h | [54] |
| $T_{G2}$ | Duration of the G2 stage of the cell cycle | 4 | h | [54] |
| $T_M$ | Duration of the M stage of the cell cycle | 2 | h | [54] |
| $R_{glycolysis}$ | Metabolic efficiency of glycolysis compared to oxidative phosphorylation | $6.67e^{-2}$ | None | [108] |
| $N_{telo,max}$ | Maximum telomere length | 60 | None | [109] |
| $N_{telo,c}$ | Critical telomere length | 20 | None | [109] |
| $P_{unrep}$ | Probability that a cell's DNA becomes unreplicated spontaneously in an hour during the S stage | 0 | None | Assumed |
| $P_{unrep,h}$ | Probability that a cell's DNA becomes unreplicated due to hypoxia in an hour during the S stage | 0.43 | None | Calibrated |
| $P_{DNA,h}$ | Probability that a cell's DNA is damaged due to hypoxia in an hour | 0.77 | None | Calibrated |
| $P_{DNA,c}$ | Probability that a cell's DNA is damaged due to chemotherapy in an hour during the S stage | 0.64 | None | Calibrated |
| $T_{apop}$ | Time scale of apoptosis | 3 | h | [110] |
| $P_{apop}$ | Probability that a cell with damaged DNA initiates apoptosis independently of CAS in an hour | 0.26 | None | Calibrated |
| $P_{apop,r}$ | Probability that a cell recovers from apoptosis in an hour | 0.96 | None | Calibrated |
| $C_{O2}^{50}$ | Oxygen concentration at which half of the living cells in the spatial domain should be necrotic | 1.2 | mmHg | [40] |
| $P_{necro,2}$ | Probability that an apoptotic cell undergoes secondary necrosis in an hour | 0.2 | None | [111] |
| $P_{necro,r}$ | Probability that a cell recovers from necrosis in an hour | 0.99 | None | Calibrated |

**Table 5. Parameters defining the neuroblastoma cell agents.** *MYCN*'s expression levels without some or all of the listed mutations of *MYCN* and *ALK* were derived from an experimental study [90].vThe wild-type MAPK/RAS pathway's signalling activity was derived from an experimental study [89].

| Parameter | Meaning | Value | Units | Source |
|---|---|---|---|---|
| $E_{p53}$ | Expression level of *p53* | 0.20 | None | Calibrated |
| $E_{MYCN}$ | Expression level of *MYCN* (*MYCN*-amplified and *ALK*-amplified or -activated) | 0.94 | None | Calibrated |
| $E_{MR,1}$ | Signalling activity of a mutated MAPK/RAS pathway (*MYCN* is amplified) | 0.38 | None | Calibrated |
| $E_{MR,2}$ | Signalling activity of a mutated MAPK/RAS pathway (*MYCN* is not amplified) | 0.00 | None | Calibrated |
| $E_{p73}$ | Expression level of *p73* | 0.14 | None | Calibrated |
| $E_{HIF}$ | Expression level of *HIF* | 0.59 | None | Calibrated |
| $E_{CHK1}$ | Expression level of *CHK1* | 1 | None | Assumed |
| $E_{p21}$ | Expression level of *p21* | 1 | None | Assumed |
| $E_{p27}$ | Expression level of *p27* | 1 | None | Assumed |
| $E_{CDC25C}$ | Expression level of *CDC25C* | 1 | None | Assumed |
| $E_{CDS1}$ | Expression level of *CDS1* | 1 | None | Assumed |
| $E_{ID2}$ | Expression level of *ID2* | 1 | None | Assumed |
| $E_{IAP2}$ | Expression level of *IAP2* | 1 | None | Assumed |
| $E_{BNIP3}$ | Expression level of *BNIP3* | 1 | None | Assumed |
| $E_{JAB1}$ | Expression level of *JAB1* | 1 | None | Assumed |
| $E_{Bcl}$ | Expression level of *Bcl-2* and *Bcl-xL*, represented as one entity in the model | 1 | None | Assumed |
| $E_{BAX/BAK}$ | Expression level of *BAX* and *BAK*, represented as one entity in the model | 1 | None | Assumed |
| $E_{CAS}$ | Expression level of *CAS* | 1 | None | Assumed |
| $E_{VEGF}$ | Expression level of *VEGF* | 1 | None | Assumed |
| $R_{diff}$ | Hourly rate of differentiation | 0.01 | None | Assumed |
| $P_{telo,r}$ | Probability that a neuroblastoma cell's telomerase or ALT pathway lengthens its telomeres in an hour | 0.09 | None | Calibrated |
| $P_{cycle,nb}$ | Probability that a neuroblastoma cell is progressing in the cell cycle at any point in time | 0.05 | None | Calibrated |

**Table 6. Parameters defining the Schwann cell agents.**

| Parameter | Meaning | Value | Units | Source |
|---|---|---|---|---|
| $P_{cycle,sc}$ | Probability that a Schwann cell is progressing in the cell cycle at any point in time | 0.03 | None | Calibrated |
| $R_{collagen}$ | Collagen production rate *per* cell | 0.71 | $\mu m^3 h^{-1}$ | [112–114] |
| $P_{DNA,r1}$ | Probability that a Schwann cell can repair its damaged DNA in an hour | 0.77 | None | Calibrated |
| $P_{DNA,r2}$ | Probability that a Schwann cell can repair its unreplicated DNA in an hour | 0.89 | None | Calibrated |

**5.6.1 Clonal competition within heterogeneous tumours.** We simulated the progression of 1200 virtual tumours with different initial clonal compositions and macroscopic properties. We used the Latin hypercube sampling technique again to generate 1200 near-random sets of initial conditions for the virtual tumours. Each set comprises the clonal composition, oxygen level, and split between the two cell populations in a virtual tumour. A clonal composition is a distribution of neuroblastoma cell agents within the population structure defined in Fig 1E. To recap, a neuroblastoma cell agent belongs to one of four clones: wild-type, *MYCN*-amplified, *TERT*-rearranged, and *ATRX*-inactivated. Each clone is in turn divided into six smaller sub-clones with six different combinations of the following mutations: *ALK* activation/amplification, other mutations activating the MAPK/RAS signalling pathway, and *p53* inactivation. Although a neuroblastoma cell agent's clonal identity is defined in terms of the subclone to which it belongs, represented by a clone ID for the sake of simplicity. The initial clonal composition of each virtual tumour was determined randomly, but only living cell agents were created. This step was executed computationally by dividing the contiguous subset of the real number line between zero and one evenly into 24 subintervals (uniform distribution), generating a random number between zero and one, and checking the subinterval it fell into. The dimensionless oxygen level was initialised randomly between $\frac{2}{72}$ and $\frac{32}{72}$ (uniform distribution); the upper and lower bounds were chosen and normalised in accordance with the literature [116]. As we were interested in malignant tumours; which have limited extracellular matrix, large populations of undifferentiated neural crest cells (subsection 5.2), and small populations of Schwann cells [95, 117]; we did not evaluate any virtual tumours with more Schwann cell agents than neuroblastoma cell agents [42]. The number of Schwann cell agents was initialised randomly between 0.05 and 0.5 of the total cell/agent count.

The following configurations were used in all 1200 simulations.

**Table 7. Parameters defining the centre-based mechanical model.**

| Parameter | Meaning | Value | Units | Source |
|---|---|---|---|---|
| $L_{overlap}$ | Overlap below which two agents cannot interact | 0 | $\mu m$ | Assumed |
| $k_1$ | Linear force law parameter | $2.20e^{-3}$ | N m$^{-1}$ | [92] |
| $L_{nghbr}$ | Search radius of an agent | 17.33 | $\mu m$ | Calibrated |
| $N_{nghbr,max}$ | Maximum number of agents allowed within an agent's search radius, including itself. Contact inhibition activates beyond this threshold | 2 | None | Assumed |
| $k_2$ | Factor by which an agent magnifies the force acting on it upon contact inhibition | 2 | None | Assumed |
| $\mu$ | Viscosity in the absence of extracellular matrix | 0.40 | N s m$^{-1}$ | [92] |
| $\Delta t$ | Time scale of cell-cell mechanical interactions | 36 | s | [92] |
| $k_3$ | Factor by which the continuous automaton can expand in each dimension | 1.26 | None | [115] |
| $k_4$ | Factor by which an agent beyond the continuous automaton's boundary is displaced back into it in each dimension | 2 | None | Assumed |

1. As we were interested in malignant tumours, a high cellularity of 0.8—precise value chosen *ad hoc*—was used, meaning that 80% of the spatial domain was accessible to the cell agents therein, while 20% was full of extracellular matrix. The neuroblastoma cell agents were initialised as undifferentiated neural crest cells (explained in subsection 5.2).

2. Although neuroblastoma diameters are known to vary from 2 cm to 20 cm [118], the virtual tumour's volume was initialised at 8 mm$^3$ due to computational limitations, but this volume corresponds to approximately $6 \times 10^5$ cells.

3. Each run lasted 3024 one-hour time steps. This time scale is compatible with rapid COJEC, the chemotherapy regimen used in the induction phase of the standard of care for high-risk neuroblastoma [3], which uses fixed doses of chemotherapeutic agents in eight two-week cycles [102, 103].

4. As we wanted to understand disease progression without treatment, therapeutic interventions were not simulated in the first study.

For each virtual tumour, the stochastic process was realised 10 times. The average final number of neuroblastoma cells in each subclone and the average final number of Schwann cells were obtained, defined as the virtual tumour's ensemble outcome. As shown in subsection 2.3, 10 repeats can give a reliable estimate of the model's average behaviour in terms of the cell counts in a simulation.

**5.6.2 MA clone's sensitivity to its gene expression profile.**   In order to find intracellular conditions favouring *MYCN*-amplified clones, we tested 1000 gene expression profiles. We used the Latin hypercube sampling technique to generate 1000 near-random gene expression profiles. Due to the high dimensionality of these gene expression profiles (20), only the expression levels of *MYCN*, the genes encoding the MAPK/RAS signalling pathway, *p53*, *p73*, and *HIF* were varied. We picked *MYCN* due to our focus on *MYCN*-amplified clones in this sensitivity analysis, the MAPK/RAS pathway due to its dependence on whether *MYCN* is amplified [89], *p53* and *p73* due to their contradictory functions of promoting cell death and DNA repair [119], and *HIF* due to its central role in the hypoxic response [120].

The neuroblastoma cell agents of every virtual tumour were initially evenly distributed between its 24 subclones. In each simulation, the neuroblastoma cell agents were assigned static gene expression levels. The study was designed to assess each *MYCN*-amplified clone's dynamics relative to the other three clones in a growing population, so the oxygen level of every virtual tumour was initialised to promote growth. An appropriate value was chosen based on the results generated from the study described in subsubsection 5.6.1: 0.24. This was obtained by averaging the initial oxygen levels of the 1155 progressing virtual tumours simulated in the first study. For the same reason and by the same method, 28% of the initial agents were set to be Schwann cell agents. The cellularity (0.8) used in the first set of simulations was retained and like the first set, the neuroblastoma cell agents were initialised as undifferentiated neural crest cells. Otherwise, the simulations in this set were configured in the same way as the first set (subsubsection 5.6.1).

For each gene expression profile, the stochastic process was realised with the same configuration 10 times and the final cell counts (neuroblastoma cells in each subclone and Schwann cells) were averaged.

**5.6.3 MA clone's sensitivity to its fractional composition and microenvironment.**   After selecting a gene expression profile that behaved in accordance with the literature [20] in the second set of simulations, we simulated the dynamics of 10 *MYCN*-amplified clones in virtual tumours with five initial fractional compositions and two oxygen levels. In these simulations, zero, 25, 50, 75, or 100% of the initial neuroblastoma cell agents in a virtual tumour were

allocated to its *MYCN*-amplified clone, while the rest were placed in its wild-type clone. Within each clone, the agents were initially distributed between the six subclones evenly. Two initial oxygen levels were used: 0.24 as in the second set (subsubsection 5.6.2) and $\frac{32}{72}$, the upper limit in the first set (subsubsection 5.6.1).

In all 10 simulations, inside every neuroblastoma cell agent, the gene expression levels were fixed in accordance with the selected profile. Otherwise, the simulations in this third set were configured in the same way as the second set (subsubsection 5.6.1).

Due to the small number of configurations in this set, we could afford more repeats *per* configuration. We tested the model's robustness by resampling the results and building confidence intervals. The stochastic process was realised 100 times *per* configuration, not 10 times as before, to facilitate resampling.

**5.6.4 Evaluation of combination therapies.**    The final set of simulations actually refers to four separate *in silico* drug trials, wherein only one clone (wild-type, *MYCN*-amplified, *TERT*-rearranged, and *ATRX*-inactivated) was present in the virtual tumour used in each trial. In each trial, we simulated the relevant clone's responses to 5000 drug combinations inhibiting the 20 gene products in each neuroblastoma cell agent. We used the Latin hypercube sampling technique to generate 5000 near-random drug combinations. A drug combination can be interpreted as a vector containing 20 probabilities. Each probability is the likelihood of inhibiting a particular gene product at any point during the relevant simulations. In addition to these four sets of 5000 simulations, we simulated a control case for each clone: its dynamics in the absence of drugs.

At the start of a simulation, all the neuroblastoma cell agents were assigned to one clone: wild-type, *MYCN*-amplified, *TERT*-rearranged, or *ATRX*-inactivated. Within the assigned clone, the agents were distributed evenly over its six subclones. An effective drug combination must work in heterogeneous microenvironments [116], but our model can only describe a small sub-region in a simplified manner. Generally, the variations in a sub-region do not reflect the variations in the tumour to which the sub-region belongs. Therefore, we wanted to test every drug combination throughout the entire range of oxygen level. To do so, the dimensionless oxygen level was initialised at 10 regular intervals between zero and one in the 10 runs making up each simulation. Malignant neuroblastic tumours are stroma-poor (limited extracellular matrix), undifferentiated, and have small populations of Schwann cells [95, 117]. At the beginning of every simulation, 11% of the agents were set to be Schwann cell agents, lower than the initial fractions used in the first three sets of simulations. In our model, this setting represents an undifferentiated neuroblastoma—the most malignant type of neuroblastic tumour with the lowest level of infiltration by Schwann cells [42].

Otherwise, the simulations in this set were configured in the same way as the third set (subsubsection 5.6.3). Although targeted therapeutic rather than chemotherapeutic agents were combined in this set, the time scale (3024 hours) used in the earlier studies was retained. In a clinical trial of crizotinib, the neuroblastoma patients received crizotinib twice a day in cycles of 28 days [121]. Conventional chemotherapy was not used in that trial. The patients were given different numbers of cycles, ranging from one to 39, but most were given fewer than five cycles (3360 hours), similar to our time scale.

The stochastic process was realised 10 times *per* configuration.

## 5.7 Hardware, software, and data analysis procedures

The model was implemented in FLAME GPU 2 (source code: link), an agent-based simulation library targeting NVIDIA graphics processing units (GPUs) through the use of CUDA [37].

The large-scale simulations were run on NVIDIA A100 GPUs provided by the University of Sheffield's High Performance Computing service.

From the first set of 1200 simulation results, 1155 virtual tumours were found to be progressing when the corresponding simulations ended. In each case, the final enrichment levels of the four major clones were extracted. Using the R programming language's *t.test* function, we performed a one-tailed paired sample t-test on the subsets associated with the wild-type and *MYCN*-amplified clones respectively, assuming that they are samples of two normal distributions with the same variance. We tested the null hypothesis that the true means of the two populations (normal distributions) are the same. In order to test whether the three subsets of final enrichment levels associated with the wild-type, *TERT*-rearranged, and *ATRX*-inactivated clones are statistically different, we performed an ANOVA analysis. The R programming language's *aovp* and *aov* functions were utilised to conduct an exact permutation test and an F-test on them respectively. They both test whether multiple population means are equal, as opposed to the t-test, which compares two means only. Our null hypothesis was that the three underlying populations have the same mean.

From the second set of 1000 simulation results—one for each gene expression profile—870 profiles were shortlisted for further consideration. Each shortlisted profile led to a higher average number of living neuroblastoma cell agents when the corresponding simulation ended than it had started with. These 870 profiles were further filtered to find the profiles that allowed the *MYCN*-amplified clones to dominate the wild-type clones in their respective simulations. Dimensionality reduction was attempted to find latent features in the resulting 283 gene expression profiles. The Python module scikit-learn (decomposition) was used for this purpose. Out of these 283 profiles, one was selected according to the literature [20]. First, we kept the profiles that allowed the three groups of mutated clones to outgrow the wild-type clones in their respective simulations. When these simulations ended, each of the three mutated clones in each virtual tumour contained more neuroblastoma cell agents than the wild-type clone therein. In other words, the mutated clones in these simulations were fitter than their wild-type counterparts, a trend consistent with the literature [20]. Out of the remaining profiles, we picked the ones that allowed the subclones with mutations in *p53* and the genes encoding the MAPK/RAS signalling pathway to outgrow their wild-type peers in their respective virtual tumours. Only five gene expression profiles remained after this step. According to the literature [20], neuroblastoma cells with amplified or activated *ALK* are less fit than those with mutations in the other genes encoding the MAPK/RAS pathway. None of the five remaining profiles led to a simulation following this trend, so the closest match was selected. It is noteworthy that this profile allowed all four clones to expand (more living cells) in the corresponding simulation. The four sets of final clone sizes (one wild-type and three mutated) corresponding to this profile were subjected to rigorous statistical tests. Using the R programming language's *t.test* function, we performed a one-tailed paired sample t-test on the final *MYCN*-amplified and wild-type clone sizes, assuming that they are samples of two normal distributions with the same variance. We tested the null hypothesis that the true means of the two populations (normal distributions) are the same. We did the same for the final *TERT*-rearranged and *ATRX*-inactivated clone sizes relative to the final wild-type clone sizes too. In order to test whether the final sizes of the three mutated clones are statistically different, we performed an ANOVA analysis. The R programming language's *aovp* and *aov* functions were utilised to conduct an exact permutation test and an F-test on them respectively. They both test whether multiple population means are equal, as opposed to the t-test, which compares two means only. Our null hypothesis was that the three underlying populations have the same mean.

The results generated from the third set of simulations were used to test the model's robustness. Depending on the initial clonal composition, a simulation led to either six or 12 datasets for this analysis: one for each subclone that was present when the simulation began. The datasets record their final sizes. In total, 96 datasets were generated. Each dataset contains 100 samples, one for each run. Using the R programming language's *sample* function, we resampled each dataset 100 times. Every time, we took 10 of its 100 samples with replacement, meaning that we took a sample/run, recorded the final size, and returned the sample to the dataset before taking the next sample. Then, we calculated the mean of the 10 recorded final sizes: the sample statistic. Using the R programming language's *CI* function, we calculated the 95% confidence interval of the 100 resulting sample statistics, as well as their own mean. This resampling method gave us 96 confidence intervals, one for each dataset.

The fourth set of simulation results were analysed to find the drug combinations that were effective in the four drug trials. Each of the four control simulations. ended with around 1.1 million living neuroblastoma cell agents. Considering each trial separately, we searched for drug combinations that led to more than 90% of this baseline (1000000 living neuroblastoma cell agents) when their respective simulations ended. We found 265–297 such ineffective drug combinations in the four trials, as well as 21–26 drug combinations that had led to regressed tumours in their corresponding simulations. In order to obtain a more balanced dataset for each trial, we relaxed the definition of an effective drug combination, including any combinations that led to fewer than 1000 living neuroblastoma cell agents when their corresponding simulations ended, finding 260–305 effective drug combinations. We merged the four groups of ineffective drug combinations with their corresponding effective drug combinations. An example in each of the four merged datasets comprises 20 inhibitory activity levels. The Python module scikit-learn (decomposition) was used to conduct a principal component analysis on each merged dataset. In each case, after projecting the data on the first principal component, we performed hierarchical clustering along that axis using Python's scikit-learn.cluster module, specifically AgglomerativeClustering with a linkage criterion of 'ward'. The examples in the dataset were merged along the first principal component to form clusters by minimising the total intra-cluster variance. After obtaining two clusters in this way, we utilised Python's sklearn.metrics module, specifically silhouette_score, to assess their compactness and distinctness on the first principal component. Generally, an example's silhouette coefficient indicates how similar it is to its own cluster (compactness) compared to how different it is from the other clusters (distinctness).

## Supporting information

**S1 Text. Detailed model description.** This text describes each model component in detail, including the continuous automaton, both agent types, the centre-based mechanical model, and the auxiliary functions linking them together in the overall multicellular model.
(PDF)

**S2 Text. Model parameterisation.** This text explains how we used results from the literature to parameterise the model.
(PDF)

**S3 Text. Implementation of the stochastic simulation process.** This text provides the details regarding each step of the simulation algorithm.
(PDF)

**S4 Text. Calibration.** This text explains how we fixed the unconstrained parameters.
(PDF)

## Author Contributions

**Conceptualization:** Kenneth Y. Wertheim, Paul Richmond, Dawn Walker.

**Data curation:** Kenneth Y. Wertheim, Robert Chisholm.

**Formal analysis:** Kenneth Y. Wertheim.

**Funding acquisition:** Paul Richmond, Dawn Walker.

**Investigation:** Kenneth Y. Wertheim.

**Methodology:** Kenneth Y. Wertheim, Paul Richmond.

**Project administration:** Dawn Walker.

**Resources:** Paul Richmond.

**Software:** Robert Chisholm, Paul Richmond.

**Supervision:** Paul Richmond, Dawn Walker.

**Validation:** Kenneth Y. Wertheim.

**Visualization:** Kenneth Y. Wertheim.

**Writing – original draft:** Kenneth Y. Wertheim.

**Writing – review & editing:** Kenneth Y. Wertheim, Robert Chisholm, Paul Richmond, Dawn Walker.

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
