## [Decision Letter · Decision Letter 0]

23 Jul 2024

Dear Dr. Walker,

Thank you very much for submitting your manuscript "Multicellular Model of Neuroblastoma Proposes Unconventional Therapy Based on Multiple Roles of p53" for consideration at PLOS Computational Biology. As with all papers reviewed by the journal, your manuscript was reviewed by members of the editorial board and by an independent reviewer. The reviewer appreciated the attention to an important topic. Based on the review, we are likely to accept this manuscript for publication, providing that you modify the manuscript according to the review recommendations.

Sincerely,

Martin Meier-Schellersheim

Academic Editor

PLOS Computational Biology

Pedro Mendes

Section Editor

PLOS Computational Biology

Reviewer's Responses to Questions

**Comments to the Authors:**

Reviewer #1: Your study introduces an impressive multicellular model of neuroblastoma, delving into the dynamics of MYCN amplification clones and exploring various chemotherapy combinations. Moreover, your efforts to validate the model's capacity to reproduce established literature findings and underscore its potential for generating digital replicas of neuroblastoma patients are admirable. Below, I have outlined some comments and suggestions aimed at further enhancing the clarity, rigor, and impact of your manuscript:

1 - In addressing the abstract and specific parts of the text, the description of the major components of the multicellular model of neuroblastoma could benefit from refinement. Emphasizing the primary features, such as the continuous automaton and the center-based agent-based model, would enhance clarity and conciseness in conveying the key aspects of the model.

2 - In certain sections of the manuscript, the model is referred to as the first computational model of neuroblastoma. However, it is important to note that this is not the initial model constructed, as portions of this model have previously been published by the authors in the article cited: https://doi.org/10.1016/j.cmpb.2023.107742. Clarifying this point would help ensure accuracy in the description of the model's novelty and previous iterations.

3- To enhance clarity and facilitate better referencing in the explanation, it is suggested to break down Figure 1a into subfigures. Additionally, it appears that part of the figure, specifically the green fluorescent marker cells, lacks explanation, and it seems disconnected from the model structure.

4- Consider rearranging Figure 3 to improve readability, particularly in areas where the letters are too small.

5- In line 146, the statement mentions, "An ensemble was categorized only if the majority of its runs had ended in one of the three outcomes." It is important to address the uncertainty surrounding this classification and whether it converges over repeated runs. Additionally, clarification is needed regarding how many ensembles have unique classifications and how the outcomes of the ensemble change in accordance with this hypothesis. Furthermore, there is a lack of commentary on why no differentiation occurs and whether any run replicate consistently yields this outcome. These observations are significant, particularly in the context of precision medicine.

6- In line 151, it is stated that "simulations. Figure 4(b) indicates that, as expected, hypoxia led to regression. On the other hand, the initial clonal composition of a virtual tumor had very little impact on its ensemble outcome (figures 5 and 6)." While this observation is notable, incorporating statistical tests into the figures would provide a more scientifically rigorous analysis rather than relying solely on visual inspection. Additionally, comparing outcomes by medians at the end of this paragraph may not be sufficient to determine if there are any significant variations. Therefore, a more comprehensive statistical analysis is warranted.

7- In line 201, the statement mentions, "It is also clear that one expression level (MR1) of the MAPK/RAS signaling pathway is upregulated relative to the other (MR0). However, MR1 was set to be higher than MR0 to reflect the influence of MYCN amplification on the pathway, so the difference is trivial." This sentence sounds more like a hypothesis than a result. Additionally, if the difference is considered trivial, it may not be necessary to include it in the results.

8- In line 204, it is stated, "Performing the same analysis on the 870 progressing virtual tumors' profiles gave us a first principal component responsible for 20% of the variance in the larger dataset (not shown)." If the information is not shown, it should not be included in the main text unless it is provided in the Supplementary Material. Additionally, at the end of this paragraph, it is asserted that there is not a latent feature. However, this conclusion requires further scientific evidence. Specifically, it is necessary to define what constitutes a small difference (as "small" is relative) and what qualifies as a latent feature in this context.

9- In line 218, it is stated that "On average (10 runs), the wild-type, MYCN-amplified, TERT-rearranged, and ATRX-inactivated clones contained approximately 2.63 × 10^5 (261873–265109), 2.68 × 10^5 (266207–269300), 2.63 × 10^5 (261921–265248), and 2.63 × 10^5 (261800–264824) living neuroblastoma cell agents when this simulation finished." To strengthen the analysis, a statistical test of these populations could confirm whether they appear identical, and whether any differences are statistically significant rather than merely minute in relative terms. Additionally, the assertion of consistency lacks scientific grounding; it would be beneficial to support this claim with evidence from the literature or provide an explanation for the potential consistency observed.

10- In line 241, it is stated, "Working with 96 datasets comprising 100 final subclone fractions each, we resampled each dataset 100 times, taking 10 samples with replacement and averaging them arithmetically to obtain a sample statistic every time, leading to 100 sample statistics per dataset." However, there is a lack of description regarding the specific nature of the 96 datasets being referenced.

11- In the Materials and Methods section, it's important to include a summary that provides an overall description of the integration of methods, including the model components such as the continuous automaton, autonomous discrete agents, and mechanical model. This summary will help readers understand how these methods are combined and utilized in the study's approach.

12- In line 634, the statement "This simplified representation of the microenvironment, especially the assumption of homogeneity, is justified as the model only describes a small region in a tumor" lacks clarification on the specific size of the region for which the hypothesis of homogeneity is valid. A literature review on the topic could provide insights into the appropriate scale at which homogeneity assumptions hold true in tumor microenvironments.

13- In line 684, the statement mentions, "If it is apoptotic or necrotic, it may be removed from the system (imitation of how immune cells act)." To clarify the observation within the parenthesis regarding how this imitation occurs, it would be helpful to provide specific details or mechanisms through which the removal of apoptotic or necrotic cells from the system is simulated in the model to mimic the behavior of immune cells.

14- In the Mechanical Model section (6.3), including a mention that the off-lattice model is a center-based model would enhance clarity and facilitate understanding for readers. Additionally, at the end of this section, it is stated, "Empirical tests were conducted..., which was judged to be physically realistic." Providing more details about the criteria used to judge the physical realism of the model, such as the magnitude of speed and typical cellular radius, would offer a more comprehensive understanding of the evaluation process.

15- In section 6.4.1, it would be beneficial to enhance clarity by listing the inputs required for the simulation before directing the reader to other paragraphs where these inputs are described. This approach can help readers understand the key components of the simulation process more effectively.

16- In line 738, it is suggested to remove the sentence "which has a smaller fraction of Schwann cell agents and less differentiated neuroblastoma cell agents than ganglioneuroma," as it appears to be duplicative.

17- In line 740, the phrase "our desire to use the whole range of each variable" does not provide a scientific rationale for the exploration. Consider revising this statement to reflect a more scientifically grounded motivation for the approach taken in the study.

18- In line 744, it's necessary to define the parameter "deg_{diff,n}" and clarify how it is stochastically determined, including the distribution used. Similarly, this information should also be provided in lines 754, 759, and 762.

19- In the model calibration section, it's essential to provide details on how the 22 parameters were chosen and the selection ranges for each parameter. This information is crucial for understanding the calibration process and ensuring transparency in the model development.

20- In some calibration studies, additional details are needed regarding the problem statement, including the target and metrics used in the calibration process.

a) In the first round, clarification is needed on how a quantity representing the growth kinetics was defined and identified as the target.

b) In the fourth round, it's necessary to define how the histology of the tumor and clinical outcomes were determined and utilized to select the four candidates from the initial ten. In the main manuscript, it is mentioned that this selection was based on the cited "realist probabilities" in line 837, but further clarification is needed on the meaning of this term and how it was quantified.

c) In the fifth and sixth rounds, information is missing about the metrics and targets used to select three from four candidates, as well as in the sixth round for selecting one from four candidates.

Providing these details will enhance transparency and understanding of the calibration process. For the other rounds, the supplemental material has a satisfactory description.

21 - In section 6.6.1:

a) It could be beneficial to define the population structure instead of requesting the reader to look at Figure 1, which may be distant from the current text.

b) Be more specific about what is meant by "determined randomly," including clarification on the distribution used (uniform, normal, etc.).

c) In the definition of the range of oxygen level, provide an explanation if these values have units associated with them.

**Have the authors made all data and (if applicable) computational code underlying the findings in their manuscript fully available?**

Reviewer #1: Yes

PLOS authors have the option to publish the peer review history of their article (what does this mean?). If published, this will include your full peer review and any attached files.

Reviewer #1: No

Figure Files:

Data Requirements:

Reproducibility:

References:

---

## [Decision Letter · Decision Letter 1]

18 Nov 2024

Dear Dr. Walker,

We are pleased to inform you that your manuscript 'Multicellular Model of Neuroblastoma Proposes Unconventional Therapy Based on Multiple Roles of p53' has been provisionally accepted for publication in PLOS Computational Biology.

Best regards,

Martin Meier-Schellersheim

Academic Editor

PLOS Computational Biology

Pedro Mendes

Section Editor

PLOS Computational Biology

Feilim Mac Gabhann

Editor-in-Chief

PLOS Computational Biology

Jason Papin

Editor-in-Chief

PLOS Computational Biology

Reviewer's Responses to Questions

**Comments to the Authors:**

Reviewer #1: All comments have been provided directly in the response letter and the revised manuscript. There are no additional attachments.

**Have the authors made all data and (if applicable) computational code underlying the findings in their manuscript fully available?**

Reviewer #1: Yes

PLOS authors have the option to publish the peer review history of their article (what does this mean?). If published, this will include your full peer review and any attached files.

Reviewer #1: No

---

## [Editor Report · Acceptance letter]

12 Dec 2024

PCOMPBIOL-D-24-00310R1 

Multicellular Model of Neuroblastoma Proposes Unconventional Therapy Based on Multiple Roles of p53

Dear Dr Walker,

I am pleased to inform you that your manuscript has been formally accepted for publication in PLOS Computational Biology. Your manuscript is now with our production department and you will be notified of the publication date in due course.

With kind regards,

Zsofia Freund
